# Importance of subsurface water for hydrological response during storms in a post-wildfire bedrock landscape

Abra Atwood [1,7] ✉, Madeline Hille [2,5,7] ✉, Marin Kristen Clark[2], Francis Rengers [3], Dimitrios Ntarlagiannis [4], Kirk Townsend [2,6] & A. Joshua West [1]

Wildfire alters the hydrologic cycle, with important implications for water supply and hazards including flooding and debris flows. In this study we use a combination of electrical resistivity and stable water isotope analyses to investigate the hydrologic response during storms in three catchments: one unburned and two burned during the 2020 Bobcat Fire in the San Gabriel Mountains, California, USA. Electrical resistivity imaging shows that in the burned catchments, rainfall infiltrated into the weathered bedrock and persisted. Stormflow isotope data indicate that the amount of mixing of surface and subsurface water during storms was similar in all catchments, despite higher streamflow post-fire. Therefore, both surface runoff and infiltration likely increased in tandem. These results suggest that the hydrologic response to storms in post-fire environments is dynamic and involves more surface-subsurface exchange than previously conceptualized, which has important implications for vegetation regrowth and post-fire landslide hazards for years following wildfire.

Wildfires can profoundly change landscapes, most obviously by their effect on vegetation, but also by altering hydrologic and geomorphologic processes. Wildfire frequency and size are expected to increase as global climate change affects seasonal temperature and precipitation intensity extremes[1–4]. More frequent and intense fires could exacerbate floods and debris flows, increase erosion, and imperil water resources[5–7]. One of the commonly observed hydrologic effects of wildfire is an increase in storm streamflow from burned areas relative to unburned areas[8–11]. In southern California, this hydrologic response has been primarily attributed to infiltration-excess surface runoff due to high rainfall rates coupled with changes in surface properties, including but not limited to hyper-dry conditions immediately after a fire[12], soil compaction from rainwater

impact[13], sealing from ash clogging[14], soil water repellency[15], and a decrease in surface roughness[16]. In association, and particularly in climates dominated by convective storms, the probability of runoff-generated debris flows increases dramatically in the immediate years following a wildfire[17,18], with these debris flows linked to changes in surface water transport.

Set against this long-standing paradigm of enhanced post-fire stormflow, recent work has indicated that the hydrologic effects of fires may be more complex, involving changes in both surface and subsurface water. The creation of macropores from burned vegetation[19,20], combined with decreased evapotranspiration following vegetation mortality, may increase groundwater storage[21–23]. In some cases, the hydrologic regime can change for years after a wildfire,

[1]Department of Earth Sciences, University of Southern California, Los Angeles, CA, USA. [2]Department of Earth and Environmental Sciences, University of Michigan Ann Arbor, Ann Arbor, MI, USA. [3]U.S. Geological Survey, Landslide Hazards Program, Golden, CO, USA. [4]Department of Earth and Environmental Sciences, Rutgers University, Newark, NJ, USA. [5]Present address: BGC Engineering, Inc., 600 12th St #300, Golden, CO, USA. [6]Present address: Exponent, Inc., 5401 McConnell Avenue, Los Angeles, CA, USA. [7]These authors contributed equally: Abra Atwood, Madeline Hille. ✉e-mail: aatwood@usc.edu; mhille@bgcengineering.ca

leading to increased baseflow[20,24–26] and aquifer recharge and desalinization[27–29]. These post-fire changes in groundwater systems observed at the annual timescale are, at least superficially, inconsistent with the expectation of increased surface runoff observed at the storm timescale. In this respect, the relationship between subsurface and surface hydrology during post-fire storms remains unclear, as does the magnitude and timescale over which the two seemingly paradoxical processes of increased surface runoff and increased groundwater storage operate[26,30]. Disentangling these aspects of the hydrologic response to wildfire, specifically during storms, is important for understanding the effect of fires on water storage, erosion, and the potential for debris flows and landslide initiation. In this work, we test whether a dynamic reservoir of subsurface moisture plays a more important role in post-wildfire storm streamflow than is often presumed in current conceptual models.

The San Gabriel Mountains, located in Los Angeles and San Bernardino Counties of southern California, USA, is an important locality for understanding post-wildfire hydrology and its effects on natural hazards and water resources[31]. Fire has long played a critical role in southern California and the San Gabriel Mountains and has modified the landscape through its effect on sediment mobility and vegetation[32,33]. Indigenous peoples of southern California used fire as a resource management technique for myriad reasons, including to increase food and material supplies and promote the growth of new vegetation[34]. During this time fires are thought to have been limited by fuel load or environmental conditions rather than suppression[35]. Currently, however, fires in the American West and elsewhere around the world are suppressed using modern equipment and techniques, leading to greater fuel loads that increase the size and intensity of fires[35]. In the San Gabriel Mountains, post-

wildfire debris flows and shallow landslides represent a substantial hazard to the surrounding urban communities, while severe droughts over the past two decades highlight the importance of understanding water resource availability in this region. Numerous studies in this area have indicated post-fire streamflow variability[24,36–38], and documented long-term wildfire effects including debris flows and shallow landsliding[39–41]. This is a region where increased streamflow post-fire is frequently attributed to reduced infiltration[40,41], but some work has also connected increased streamflow responses to increased groundwater storage[36].

In this study, we explore post-fire hydrological dynamics in the bedrock landscapes of the San Gabriel Mountains. We characterize how subsurface water storage changes post-fire, and how these changes affect streamflow and its sources[42] using a unique combination of time-lapse electrical resistivity imaging (ERI) and water stable isotope ($\delta^2$H, $\delta^{18}$O) data from storm events over two water years. Combining these methods allows us to investigate the connections between increased overland flow and increased groundwater storage. Our study is based on paired catchments, following a long-established and widely adopted approach in hydrology (e.g., Bates, 1921[43]; Bosch and Hewlett, 1982[44]). We focus on adjacent burned (Louise, ~0.07 km²; Thelma, ~0.06 km²) and mostly unburned (Henry; ~0.10 km²) catchments in the San Gabriel Mountains, following the September-December 2020 Bobcat Fire (Fig. 1). These study catchments were selected because they have broadly similar lithology, aspect, and slope steepness, while contrasting in burn intensity (Fig. 1; Fig. S1; Table S1). We find that both surface runoff and subsurface water storage are greater in the burned catchments than the unburned catchment within the two water years following the Bobcat Fire, and we posit that

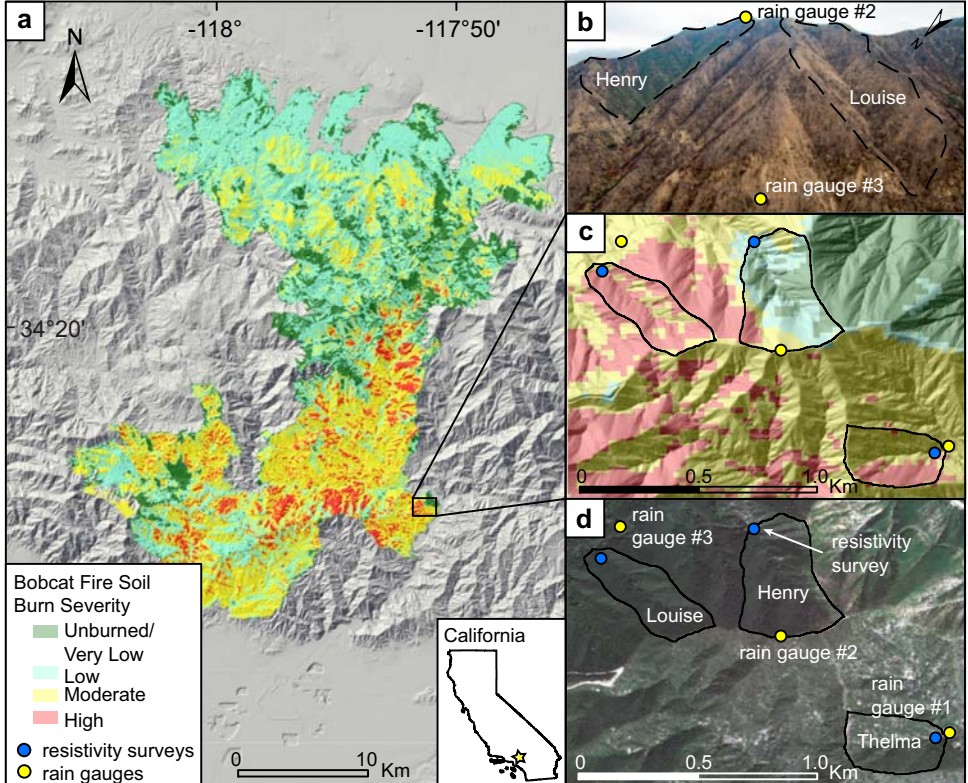

**Fig. 1 | Map and imagery overview of the study area. a** Map of the 2020 Bobcat Fire in the San Gabriel Mountains with soil burn severity (USDA Forest Service, 2020), showing the location of the study catchments. **b** Oblique photo looking south at burned and unburned catchments (photo courtesy of A.J. West). **c** Soil burn severity map of the study catchments. **d** Pre-fire imagery annotated with

locations of study catchments, rain gauges, and geophysical surveys. Pre-fire imagery was obtained by Pléiades ©CNES (2020), Distribution AIRBUSDS, sourced via SkyWatch Space Applications Inc. Lidar is from the USGS 3D Elevation Program (3DEP).

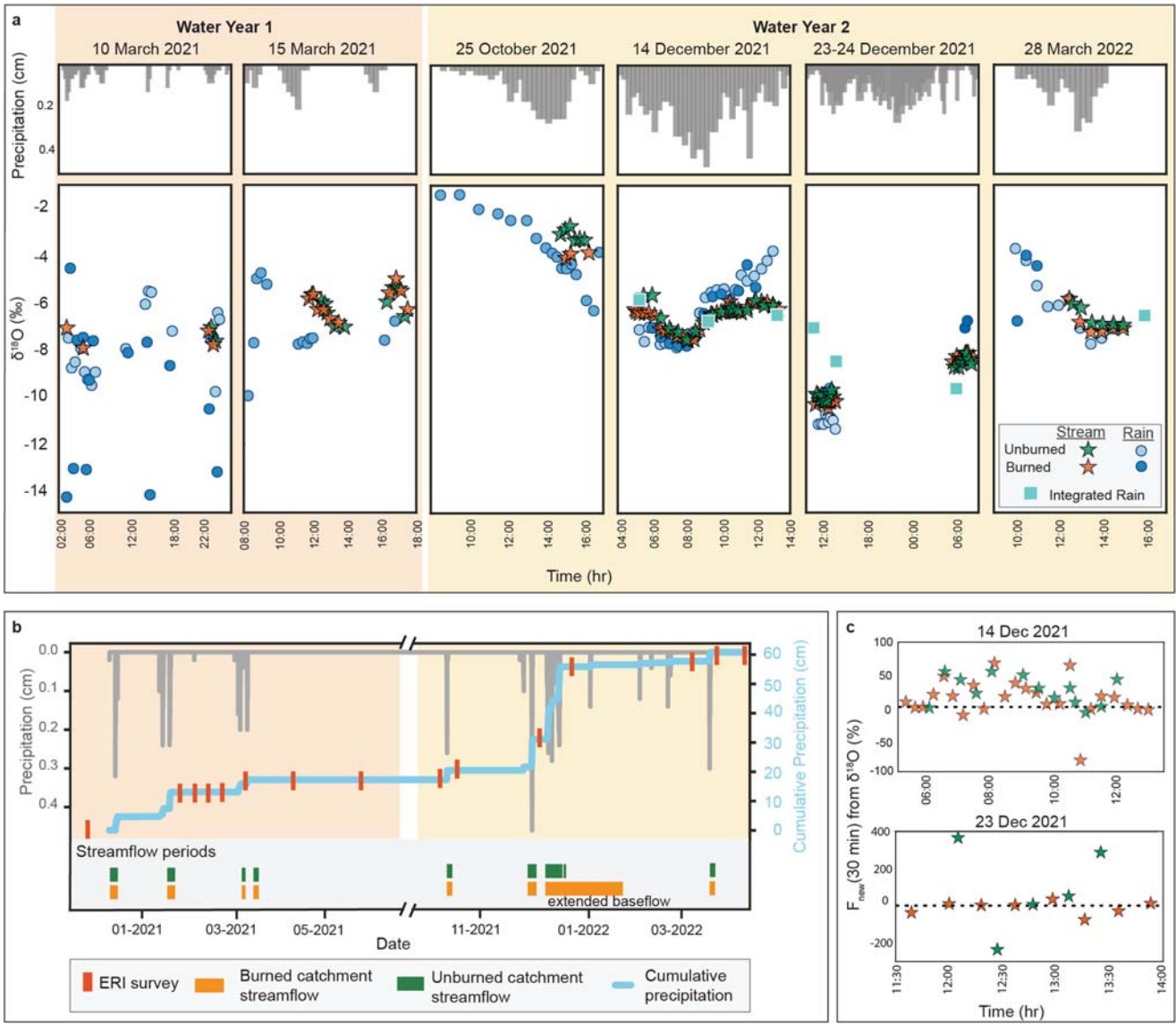

**Fig. 2 | Precipitation and streamflow data collected over the study period.**
**a** Time series of 15-minute binned precipitation and $\delta^{18}O$ from precipitation and streamflow samples during storms. Stream and precipitation samples from 10 March 2021, 14 December 2021, and 28 March 2022 are similar in isotopic composition, indicating that runoff during that event was likely generated in large part from overland flow. 15 March 2021 stream samples in both catchments and 23-24 December 2021 stream samples in the unburned catchment show a distinct signature from co-temporal precipitation. **b** Total precipitation record in 15-min binned intervals (grey bars) and cumulative precipitation (blue line) for the two water years with timing of ERI surveys (red lines). Approximate periods of streamflow are indicated by green and orange bars based on trail camera and field observations. Extended baseflow occurred in the burned catchment after the large storms in December 2021. **c** 30-min $F_{new}$ values for 14 and 23 December 2021 showing different streamflow responses. The 14 December 2021 stream response shows fluctuating amounts of new water in streamflow, whereas 23 December 2021 shows a lower fraction new water, especially in the burned catchment, indicating higher groundwater contributions.

increased post-fire streamflow is a result of dynamic connections between subsurface and surface water. Our findings add to the conceptual understanding of the hydrologic response to storms in post-fire bedrock catchments and have implications for the timescale of vegetation recovery, water resource availability, and natural hazards such as debris flows and flooding.

## Results

### Precipitation data, streamflow, and hydraulic conductivity

Nine storm events occurred during our study period from December 2020 to March 2022, with maximum rainfall intensities ranging from 7 to 20 mm/hr. Water year 1 (WY1) (October 2020-September 2021) was drier, recording ~18 cm cumulative precipitation, whereas WY2 (October 2021-September 2022) recorded ~40 cm, the majority during three major events in December 2021 (Fig. 2b). Average annual precipitation from 1990-2020 in this location was ~70 cm[45], with notable seasonal cyclicity, although this changed periodically based on El Niño/La Niña cycles.

The median values of soil surface field-saturated hydraulic conductivity ($K_{fs}$) measured post-rainfall were similar in all three catchments (Fig. 3d; Table S2). Rainfall intensity during storms rarely exceeded the median $K_{fs}$ values (Fig. S2), yet always exceeded the minimum measured $K_{fs}$ during storms where streamflow was generated. The burned catchments (Thelma, Louise) had more variable $K_{fs}$ (significant at the 5% level, based on an F-test; Table S3) than the unburned catchment, indicating substantial spatial variability in potential infiltration-excess surface runoff and infiltration.

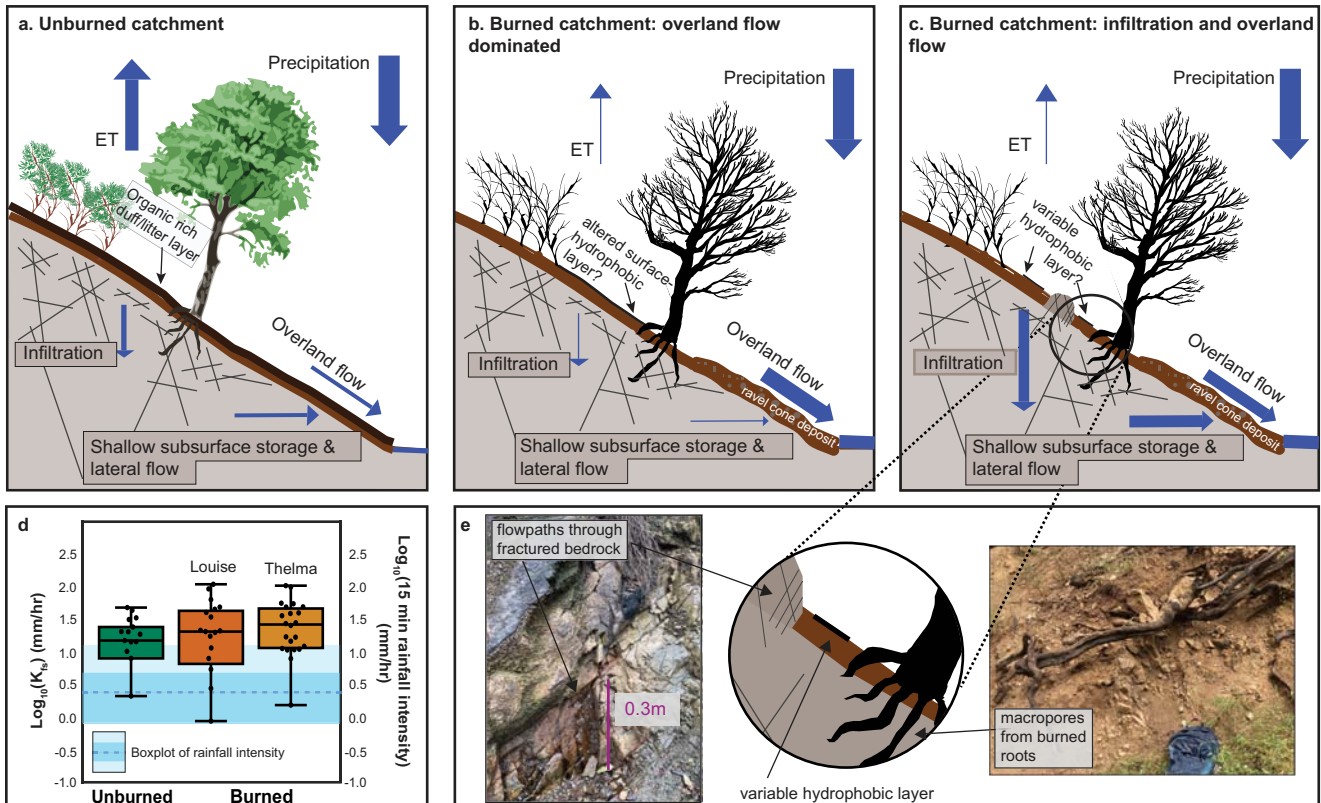

**Fig. 3 | Conceptual diagrams of the water budget. a** A typical unburned catchment, **b** a burned catchment dominated by overland flow due to hydrophobic layer or decreased surface roughness, and **c** a burned catchment with increased infiltration and surface runoff. **d** Boxplots and swarm plots of field-saturated soil hydraulic conductivity ($K_{fs}$) values from the unburned and burned catchments compared to rainfall intensity, where the points represent data values, the box represents the quartiles with the middle line indicates the median value, and the whiskers represent the data distribution excluding outliers. **e** Inset of (**c**) showing potential rapid infiltration points in burned catchments as well as photographs from the Louise catchment (left photo shows water flowing out of fractured bedrock from 4 February 2022; photos courtesy of A. Atwood). Diagrams after Jung et al.[36].

## Erosion and the subsurface characteristics

The storms in December 2021 resulted in substantial erosion and channel lowering (~2 m of channel incision at one of the burned catchments, Louise; Fig. S3). A major drawback to studying post-fire catchments is the difficulty of collecting streamflow data due to flashy, debris-filled streamflow and instrument loss, which prevented us from collecting stage height records. Nonetheless, other observations constrained streamflow dynamics for the burned-unburned pair of Louise and Henry (see Methods). Streamflow duration was broadly similar in the two catchments, except for the prolonged flow after the December 2021 storms in the burned catchment (Fig. 2b). The more pronounced difference was that the burned catchment exhibited greater streamflow during each event (Fig. S4). Calculated maximum streamflow from the December 2021 storms, inferred from channel geometry and high-flow markers near the channel outlet (Fig. S5), were 3.3 m³/s in the burned catchment and 0.51 m³/s in the unburned: ~6x higher in the burned catchment.

## Electrical resistivity imaging

ERI results allowed us to compare shallow groundwater retention patterns in burned and unburned catchments, and between WY1 and WY2. In WY1, a pulse of decreased resistivity followed the first large rainstorm in all three catchments (Fig. S6), indicating the addition of water to the subsurface. In the burned catchments, this water persisted within both the shallow and deeper subsurface (between ~1 and 7 m depth) for the remainder of WY1, even after a ~2-month period of little to no rainfall (Fig. 4b, c, k, l; Fig. 2b). Minor changes in drying and wetting at the very near surface were observed, but the subsurface

remained remarkably static in the burned catchments prior to June 2021. In the unburned catchment, water addition at the near surface (1–2 m depth) steadily dried out, and by June 2021 conditions appeared similar to those observed immediately post-fire (Fig. 4a, j).

After the dry summer months and at the start of WY2, the subsurface of the burned catchments had dried out but did not return to the immediate post-fire dryness evident at the beginning of WY1, and moisture was still evident in the near surface (between ~1–5m) (Fig. 4e, f). By comparison, the unburned catchment appeared drier at the beginning of WY2 than it did at the start of WY1, which we posit may be related to a notably low rainfall water year coupled with the effects of evapotranspiration.

Following the December and January storms of WY2, we observed a deeper wetting front within all three catchments (Fig. 4g–i) when compared to WY1, which persisted through the remainder of WY2 in both burned and unburned catchments. Over the course of the water year, near-surface drying co-located with vegetation regrowth in the burned catchments was evident more than in WY1 (Fig. 4n). Even so, in the unburned catchment, the lateral extent of drying of the near surface exceeded that of the burned catchment measured at that time (Fig. 4m).

## Storm water stable isotopes

One burned catchment (Louise) and one unburned catchment (Henry) were sampled for analysis of the stable isotope (H and O isotope) composition of rain and stream water isotopes. Precipitation $\delta^{18}O$ values ($n = 126$), which were similar between catchments, were heaviest at the beginning of WY2 and became progressively lighter into

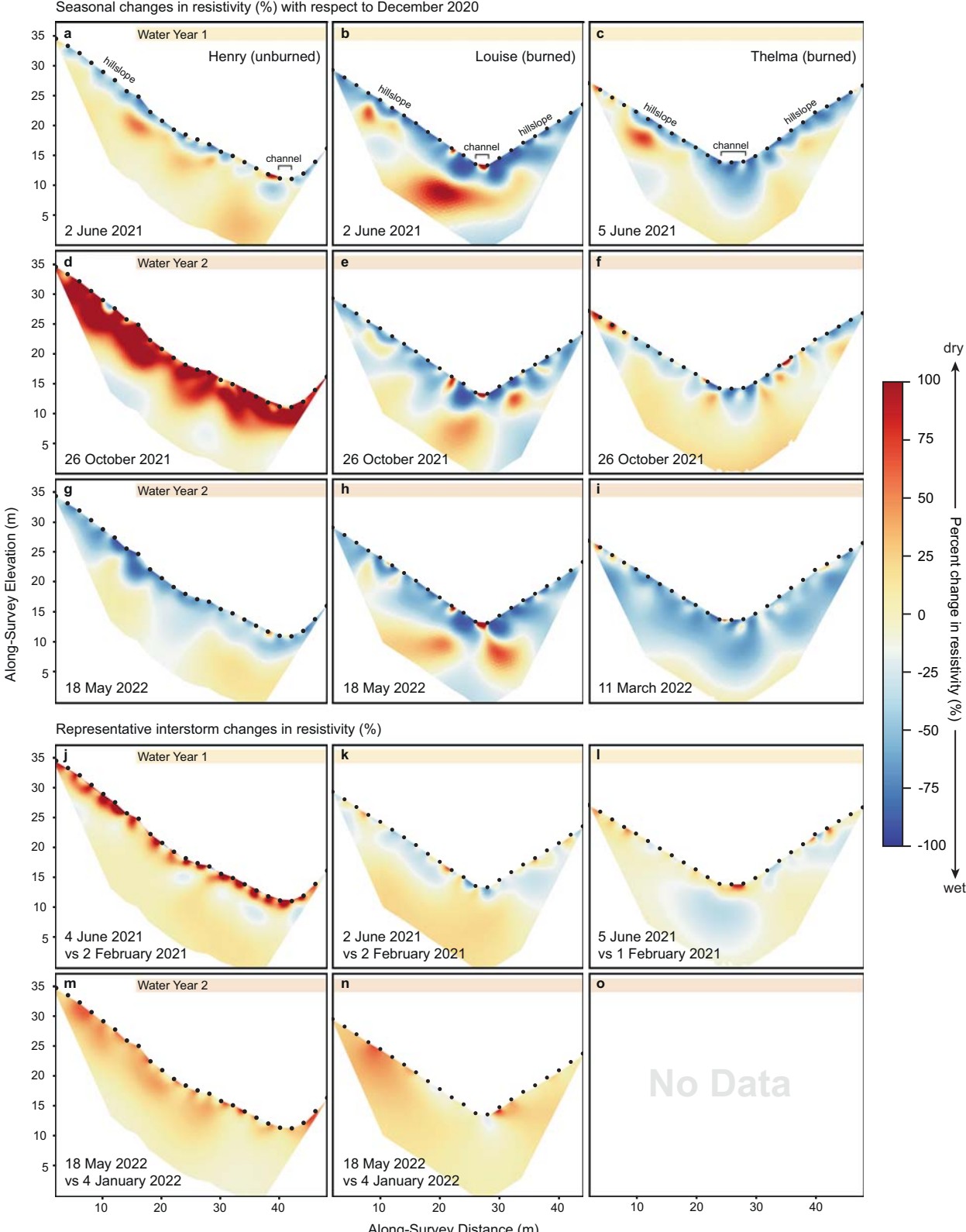

**Fig. 4 | Time-lapse electrical resistivity images at unburned (Henry) and burned (Louise and Thelma) catchments.** Images illustrating seasonal changes and interstorm changes. Seasonal time-lapse images in panels (**a**–**i**) are in reference to the baseline December 2020 surveys. Interstorm comparisons in panels (**j**–**o**) are labeled with the survey dates. Both catchments showed seasonal fluctuations of subsurface water over the course of WY1 and WY2. The unburned catchment showed shallow surface drying with resistivity exceeding early season values by the beginning of WY2 (**d**, **j**), consistent with patterns in evapotranspiration and seasonal water fluctuations after a drier-than-average year. The burned catchments showed little to no change in resistivity during WY1, indicating persistent water in the subsurface, but near-surface drying towards the late season of WY2 (**n**), which may be explained by vegetation regrowth and an increase in ET (Figs. S11 and S12).

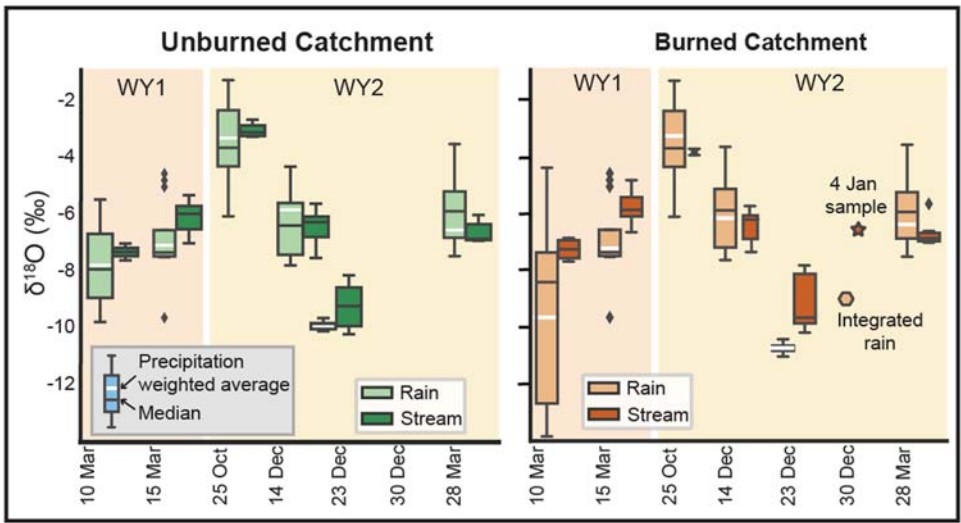

**Fig. 5 | Boxplots of δ¹⁸O measurements from rainfall and stream during storm events.** White horizontal lines indicate precipitation-weighted averages of each rainfall event (see Supplementary Text S3 for methodology), where the box represents the quartiles with the middle line indicating the median value, and the whiskers represent the data distribution excluding outliers. No time series sampling occurred during the 30 December 2021 storm, but integrated rains samples were collected and streamflow in the burned catchment from the extended baseflow period was collected on 4 January 2022. There was no streamflow in the unburned catchment to sample at that time.

January before becoming heavier again in March, following an expected sinusoidal seasonal isotope pattern[46] (Figs. 4 and 5). Precipitation δ¹⁸O composition fell along a local meteoric water line, based on δD and δ¹⁸O, with a slope of 8.3207 and intercept of 19.657 (Fig. S7). Precipitation-amount weighted average isotope compositions are presented in Table S4.

Rainwater δ¹⁸O values also showed intra-storm variability (-2–5‰) (Figs. 4 and 5). The 10 March 2021 storm, which was the storm with the least total rainfall, had the highest variability in rainwater δ¹⁸O. However, trends in δ¹⁸O values with time through storms were not always the same. In the 25 October 2021 storm, rainfall δ¹⁸O became progressively lighter with increasing precipitation, whereas in the 14 December 2021 storm, δ¹⁸O was initially lighter and then heavier as the precipitation increased. We saw some variability in δ¹⁸O values between catchments (largest on 10 March 2020), but most storms had similar δ¹⁸O values. The observed variability with time and between catchments is consistent with convective storms and spatial heterogeneity over steep mountainous topography such as that of the San Gabriel Mountains[47].

Streamflow δ¹⁸O values (n = 142) were generally less variable than precipitation δ¹⁸O values during storms and were similar between catchments (Figs. 4i, 5). However, during individual storms, we found that streamflow δ¹⁸O values could differ from contemporaneous precipitation δ¹⁸O values (Fig. 5). For example, on 10 March 2021, 25 October 2021, 14 December 2021, and 28 March 2022, the streamflow and precipitation samples had similar δ¹⁸O distributions based on overlapping mean and quartile distributions for storm totals, whereas the 15 March 2021 and 23 December 2021 streamflow samples, as well as a streamflow sample collected after the 30 December 2021 storm, had offset distributions from rainfall (Fig. 5). Intra-storm δ¹⁸O time series also reflected this pattern, where streamflow on 10 March 2021 and 14 December 2021 closely followed changes in precipitation δ¹⁸O, whereas streamflow on 15 March 2021 and 23 December 2021 did not (Fig. 2). 25 October 2021 and 29 March 2022 storms show offset precipitation and streamflow δ¹⁸O values, suggesting a delayed streamflow isotopic response to rainfall. Streamflow isotopes at the start and end of the 14 December 2021 storm deviated from precipitation values during times of lighter rainfall. These differences were also reflected in calculated D-excess values (Fig. S7).

Streamflow generated by overland flow is expected to reflect the most recent rainfall isotope signature. The fraction of new water (recent precipitation) in streamflow, which can also be thought of as the fraction from overland flow, ($F_{new}$; following Kirchner, 2019[48], Eq. 8) is a hydrograph separation technique that does not require a stationary end member and therefore can be used with changing rainfall signatures during storms. New in this context is defined by the sampling frequency. Here, we sampled every 20–30 minutes, which was the highest possible frequency for sampling given field conditions. $F_{new}$ values were calculated for 14 and 23 December 2021 storms when continuous precipitation and streamflow samples were collected to calculate $F_{new}$ at the single-storm timescale (Fig. 2c). Both catchments showed similar $F_{new}$ values during the 14 December 2021 storm, with changing $F_{new}$ values between 0-75% new water and the highest values recorded during highest precipitation time periods. During the 23 December 2021 storm, the burned catchment showed low $F_{new}$ values, while the unburned catchment had a more variable $F_{new}$ signature (Fig. 2c). In the burned and unburned catchments during both December 2021 storms, the calculated fraction of new water in streamflow samples correlated with rainfall intensities that matched or exceeded median $K_{fs}$ (Fig. S8).

## Discussion

Time-lapse ERI revealed seasonal-scale resistivity changes resulting from subsurface water accumulation and retention in the burned watersheds, indicating greater groundwater storage in this dry, bedrock environment post-fire compared with the unburned watershed (Fig. 4). Streamflow observations (Figs. S4 and S5) and water isotope data (Fig. 2) indicate that this subsurface water reservoir is dynamic and contributes to surface flow, which challenges the longstanding paradigm that in southern California, post-fire storm streamflow originates primarily from infiltration-excess overland flow.

In unburned environments, streamflow during storms is typically comprised of a mixture of rainfall and stored water, whereas stored water is any water within the catchment subsurface that becomes mobilized during storms[49–51]. In Mediterranean climates such as the San Gabriel Mountains, between 22% and 100% of streamflow can be attributed to rainfall[50], depending in part on antecedent rainfall and soil moisture conditions. Isotopic

similarities in streamflow between the unburned catchment and the sampled burned catchment in our study show that the same proportions of stored water and rainfall are driving streamflow in the two catchments (Figs. 4i, 5), despite greater observed streamflow in the burned catchment during storms (Fig. S4). Moreover, storms following an extended dry period (~2 months) showed higher proportions of surface runoff contributing to streamflow, whereas streamflow in closely spaced storms (within ~1 week) showed a relatively higher proportion of subsurface flow (Fig. 5; Fig. 2c). This observation is consistent with transient changes in subsurface storage between storms and may also reflect increased soil-water repellency and enhanced infiltration-excess surface runoff due to a reduction in soil moisture between storms.

These observations pose an apparent conundrum: how can storm streamflow be increased in the burned watershed, yet comprise a similar mixture of rainfall and stored water as in the unburned watershed? Rainfall intensities exceeded minimum $K_{fs}$ values in both catchments during storms that generated streamflow (Fig. 3d; Fig. S2), indicating that infiltration-excess overland flow was contributing to streamflow in both catchments (see further discussion in the Supplement). Yet our data indicate that a substantial portion of precipitation also infiltrated into the subsurface and, importantly, that some of this stored water was then mobilized back as streamflow during storms (Fig. 3c). We suggest that the streamflow response in the burned catchment reflects significantly enhanced subsurface exchange relative to the unburned. Our resistivity data are consistent with this interpretation of the isotopic data, with the deeper wetting front within the burned catchment indicative of overall greater water content in the subsurface compared to the unburned catchment. This subsurface water provides a larger reservoir stored in the burned catchment that can be mobilized during storms. These observations potentially reconcile differences in the study catchments' relative streamflow with the similar proportions of subsurface and surface water contributions to streamflow that we observe in the isotope data.

Although both the unburned and burned catchments experienced fluctuations in subsurface water following storms, the burned catchments exhibited more pronounced decreases in resistivity at greater depths and perhaps most importantly, showed persistent subsurface moisture over the duration of the study period, similar to observations after fire in pine forests in Texas, USA[27]. In the burned catchments in our study, water persisted at the near surface between WYs, but towards the end of WY2, drying was co-located with observed vegetation regrowth. In the unburned catchment, persistent water storage was not evident, and instead, resistivity in the near surface progressively increased as the wet season tapered and substantial drying occurred over the summer months prior to WY2, indicating drying that is consistent with observations of evapotranspiration (ET) from remote sensing data (Figs. S9 and S10).

Changes in ET over time align with patterns in resistivity; although all three catchments showed similar ET prior to the Bobcat Fire, the unburned catchment had higher ET during WY1 when vegetation regrowth in the burned catchments was limited. In WY2, ET rates rebounded in the burned catchments, indicating substantial increase in vegetation growth that is consistent with near-surface changes in resistivity in WY2 (Fig. 4n). We observe higher discharge in the burned catchments than the unburned catchment in WY2 despite this increase in ET, which we attribute to the subsurface water that persisted from WY1 to WY2 contributing to streamflow. These broad patterns in changing resistivity and ET indicate substantial, relatively deep, and lasting changes in subsurface water in the burned catchments. This change resulted in greater volumes of streamflow mobilized from the subsurface during storms, generating higher streamflow as a mixture of stored water and rainfall similar to unburned settings (Fig. 2a, c).

The differences we observe in resistivity between the burned and unburned catchments may reflect the importance of vegetation on water storage and transport in the San Gabriel Mountains. In this highly fractured and dry environment, the groundwater table is likely meters below the depth limit of the ERI surveys. Our observations document changes in the water reservoirs of fractured bedrock that fluctuate over weeks to months with the addition of water through rain infiltration and vegetation modulation. This kind of subsurface reservoir that is increasingly recognized for its hydrologic importance[52,53]. In the unburned catchment, we attribute seasonally fluctuating subsurface moisture to deeply rooted chaparral plants that hold shallow water nearer to the surface[54] (Fig. 3; Fig. 4d, j, m). In comparison, deeper and persistent water in the burned catchments may be explained by rapid infiltration of rainwater through flow paths following preferential finger flow or created by burned-out root systems or fractures exposed from the removal of the organic duff layer[19,20,27] (Fig. 3e), as well as substantial loss of reduction in evapotranspiration during WY1[55,56]. Channel erosion may have allowed the subsurface water to rejoin the surface water flow via exfiltration above channel banks.

An important implication of our observations is that a deep reservoir of water plays a role in hydrology at different timescales, from storms to seasonal and multi-year processes. Enhanced surface runoff is not the only mechanism driving increased discharge after wildfire, and the proportions of new and subsurface water in streamflow appear to be influenced by antecedent storms. We find increased subsurface storage immediately following a single storm as well as prolonged streamflow after substantial rainfall (Fig. 2b), consistent with observations of long-term hydrological effects of wildfire that include increased baseflow and aquifer recharge[25,27,28,36,57–60]. This recharge may contribute to water resources but potentially at the cost of water quality through the delivery of fire-associated contaminants including organics and heavy metals[61–64]. Greater subsurface water storage and exchange may also contribute to rock moisture storage, which is shown to buffer forests against drought conditions[52].

In addition to affecting streamflow and groundwater resources, the persistent and dynamic subsurface water storage we observe may have important implications for post-fire landscape evolution. Surface runoff is critical for the triggering of debris flows in the months following fires[33,65,66], and wet sediment may be more easily mobilized than dry sediment[67]. Our results suggest that debris flow predictions may be improved by considering spatially heterogeneous infiltration (e.g., McGuire et al., 2018[66]) and subsurface contributions to streamflow, particularly during rapidly sequential storms at the beginning of the wet season when the likelihood of a debris flow occurring is considered highest[39]. The accumulation of subsurface moisture via high infiltration pathways may also contribute to shallow landsliding through excessive pore-water pressures several years post-fire, as documented by the shift from runoff-generated debris flows to infiltration-generated debris flows in the 2–4 years following wildfire[39–41,68,69]. Shallow water storage in our study site appears to facilitate rapid vegetation regrowth, affecting post-fire ecology and potentially stabilizing hillslopes in the process[29,70]. The first-year triggering conditions for post-wildfire runoff-generated debris flows in southern California are closely tied to bursts of high-intensity rainfall, and, unlike shallow landslides, debris flow initiation is poorly correlated with soil water content[33,71]. Observations in this study and others (e.g., McGuire et al., 2018[66]) showing spatially heterogeneous infiltration can account for the fact that water can both infiltrate deeply in high-infiltration areas and run off in low-infiltration areas.

Our work demonstrates that following a wildfire, dynamic subsurface reservoirs can increase streamflow and provide a water resource for vegetation regrowth. As wildfire frequency increases in southern California and other locations, it will become increasingly important to consider subsurface hydrology in the context of post-fire

cascading hazards (e.g., shallow landslides, flooding) and ecosystem recovery.

## Methods

### Study area

We identified three first-order catchments within the perimeter of the 2020 Bobcat Fire in the San Gabriel Mountains (Fig. 1; Louise, Thelma, and Henry). All are underlain by Precambrian gneiss and Cretaceous quartz diorite of the San Gabriel Mountains range[72] and have similar slope angles, catchment areas, and long-term rainfall volume and intensity (Supplementary Information; Fig. S1; Table S1). All catchments have ephemeral streams that generate streamflow only during heavy precipitation events. During the Bobcat Fire of 2020, Louise and Thelma (burned) were moderately to severely burned over the entire area while 23.6% of Henry (unburned) was burned at moderate to high severity, with the rest unburned (Fig. 1). The comparison of these catchments allows us to study the role of different vegetation and burn severity while controlling for other major factors expected to influence hydrology and erosion.

### Precipitation data, soil infiltrometry and streamflow estimates

The climate of the San Gabriel Mountains is Mediterranean, with most precipitation falling in winter months and only sporadic rain in summer, often with months of uninterrupted dry weather[73]. We installed three tipping bucket precipitation gauges in December 2020. One of the gauges was placed near the catchment mouth of Louise (burned), the second at the top of the ridge on the south side of Louise (burned) and Henry (unburned), and the third on the opposite and southern side of the ridge at Thelma (burned; Fig. 1) to capture local-scale spatial variability and accurate timing of storm arrivals. Gauges showed similar rainfall values during WY1 (Supplementary Information); the gauge data from the catchment mouth of Louise (burned) is presented in this study.

Field-saturated hydraulic conductivity ($K_{fs}$) measurements of the catchments were made in April and August 2021 (post-rainfall), using a Meter Environmental minidisk portable tension infiltrometer with a suction of 1 cm, with multiple (~20) measurements made at each site (Table S2). The volume of water infiltrated was recorded as a function of time and converted to $K_{fs}$ using the differential linearization method[74,75]. Note that $K_{fs}$ was not measured in the upper reaches of the watersheds due to accessibility. A portion of Henry burned at moderate to high severity (~23%), which may have influenced surface runoff contributions to streamflow, but is not accounted for in the $K_{fs}$ data.

Time-lapse trail cameras were installed at the base of each catchment to capture the length of potential stream flow or erosional processes, such as debris flows. The ephemeral nature of the catchments, substantial sediment mobility during storms, and high risk of post-fire debris flows prevented installation of stream gauges to monitor streamflow.

Maximum streamflow estimates are based on channel profiles collected in March 2022. Channel profiles were collected using an RTK Septentrio Global Navigation Satellite Systems (GNSS) instrument (Fig. S9), and the cross-sectional area and the wetted perimeter were calculated using channel geometry (Fig. S5). One profile was collected above small concrete walls at the mouth of each catchment (Fig. S5) and the Froude number is 1 assuming critical flow. Maximum streamflow estimates are thus calculated according to Eq. (1):

$$V = \sqrt{gR} \qquad (1)$$

Where $V$ = velocity and $R$ = hydraulic radius of the measured channel cross-section, and streamflow ($Q$) is calculated using the cross-sectional area of the channel ($A$) using Eq. (2):

$$Q = VA \qquad (2)$$

### Electrical resistivity imaging

We collected resistivity data over 42-, 46-, and 46-m survey lengths perpendicular to the catchment channels to capture water fluctuation in both hillslopes and channels at Louise (burned), Thelma (burned), and Henry (unburned), respectively (Figs. 1, 4; Table S5; Fig. S9). We used a Syscal Pro electrical resistivity meter from Iris Instruments and a combined Dipole-Dipole and Wenner-Schlumberger array with 2-m spacing for high signal-to-noise ratio and lateral and vertical resolution[76]. Reciprocal measurements were also made at 1/3 density for robust error quantification to improve model results. We inverted two-dimensional (2D) resistivity models using ResIPy[77], an open-source inversion software[78]. For time-lapse models, we calculated changes between surveys as a percent difference in resistivity, which could reflect the addition or removal of water.

### Precipitation and streamflow collection and stable isotope analysis

Water samples were collected from the study catchments during streamflow episodes to determine δD (D/H, ‰), δ¹⁸O (¹⁸O/¹⁶O, ‰), and deuterium (D) excess (defined as Dxs = δD − 8 × δ¹⁸O, in ‰), all reported relative to Vienna Standard Mean Ocean Water (V-SMOW). Stable isotope analysis was done on Picarro L2130i cavity ringdown spectrometers at Chapman University in Orange, California, and the University of Southern California in Los Angeles, California. The internal error of isotope measurements on the Chapman spectrometer was 0.1‰ or better for δ¹⁸O and 2‰ or better for δD. The standard deviation of an independent quality control standard used for analysis at the University of Southern California was ≤0.2‰ δ¹⁸O and ≤ 2‰ for δD. Precipitation ($n = 126$) and streamflow ($n = 142$) samples were collected at high frequency (sub-hourly) during the storms on 10 and 15 March 2021, 25 October 2021, and 14, 23, and 24 December 2021. We did not sample during storms in December 2020 and January 2021 due to significant debris flow risk and associated road closures.

Precipitation samples were collected in open containers at channel outlets and transferred to 12 mL glass exetainers. During the 14 December, 23 December, 28 December 2021, and 28 March 2022 storms, buckets with mineral oil were used to collect integrated rain samples ($n = 8$) for the duration of the storm. Although variations in isotope composition due to elevation changes across the catchment are expected, past studies in this region indicate only 0.5‰ change per 500-m elevation change[79], which is consistent with data collected from integrated rain samples on the ridge and base of the burned catchment during the 28 March 2022 storm. Streamflow grab samples were collected directly into 10-mL glass exetainers, filtered using 0.2-m nylon syringe membrane and stored at 5 °C until analysis.

### $F_{new}$ calculations

New fraction water ($F_{new}$) from Kirchner (2019)[48] is a type of hydrograph separation based on correlations between tracer fluctuations in streamflow and end members, thus allowing for endmembers to change with each timestep and estimation of the amount of new water between one timestep and the next. In environmental regimes like post-fire areas where runoff is assumed to rapidly contribute to streamflow, we would expect to see high $F_{new}$ values relative to an unburned area. $F_{new}$ was calculated using Equation 8 from Kirchner (2019)[48]:

$$F_{newj} = \left( \frac{C_{Qj} - C_{Qj-1}}{C_{newj} - C_{Qj-1}} \right), \qquad (3)$$

where $C_{Qj}$ is the δ¹⁸O value of the stream water from the current timestep, $C_{Qj-1}$ is the δ¹⁸O value of the stream water in the previous timestep, $F_{newj}$ is the fraction of new water in the stream water in the current timestep, and $C_{newj}$ is the δ¹⁸O value of the new precipitation that fell during the current timestep.

## Data availability
The data used in this study are available in the Hydroshare database under the Creative Commons Attribution CC BY at http://www.hydroshare.org/resource/670e332937c148eb94178f0e4e18cdd7.

## Code availability
The code used in this study is available at the following Github repository: https://zenodo.org/badge/latestdoi/595292040.

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

## Acknowledgements
We thank B. Atwood, A. Lunstrum, C. Chen, T. Callahan, S. LaHusen, N. Niemi, W. Medwedeff, P. Blunts, S. Keating, R. Kelly, I. Smith, E. Burt, R. Rumaga-Montenegro, C. Rosen, K. Barnhart and J. Kostelnik for help with fieldwork. We also thank Ann Berkley of the U.S. Forest Service for help with access and permitting. A.J.W. and A.A. disclose support for this research of this work by NSF-FRES #2021619. F. Rengers discloses support for this research of this work by the U.S. Geological Survey Landslide Hazards Program. M.K.C discloses support for this research of this work by NSF-FRES #2020970 and NASA DISASTERS #80NSSC20K1032. Any use of trade, firm, or product names is for descriptive purposes only and does not imply endorsement by the U.S. Government.

## Author contributions
A.A., M.K.C., and A.J.W. designed research; A.A., M.H., A.J.W., M.K.C., F.R., and K.T. performed research; D.N. contributed analytical expertise; A.A., M.H., F.R., and D.N. analyzed data with contributions from M.K.C.; A.A. and M.H. wrote the paper with feedback from all other authors.

## Competing interests
The authors declare no competing interests.
