## [Peer Review File · Nature Communications]

Importance of subsurface water to hydrological response during storms in a post-wildfire bedrock landscapeREVIEWER COMMENTS

Reviewer #1 (Remarks to the Author):

This manuscript evaluates subsurface moisture dynamics following fire in a chaparral system and compares a severely burned catchment with a nearby unburned catchment using electrical resistivity imaging and stable isotopic analysis. Much of the previously published research on fire effects to hydrology has been focused on surface runoff generated from reduced infiltration rates often attributed to hydrophobic soils and until recently relatively few papers have considered subsurface effects from wildland fire. This paper does an admirable job conceptualizing how fire affects will change subsurface moisture conditions, which in turn is reflected in catchment discharge magnitude and duration. This conceptualization is supported in this manuscript using a paired catchments and detailed resistivity and isotopic observations. The manuscript is well written, and I find the methods used to be robust. It is my opinion that this paper deserves publication as it adds to the growing research that describes coupled surface and subsurface hydrology considering fire disturbance. As a result, most of my comments are editorial in nature with the aim to improve the paper. I will say that publishing in Nature Communication deserves a level of scientific impact and novelty that needs to be evident. Here the authors could improve this aspect by identifying which processes and concepts will apply or add to knowledge outside of the Chaparral system as well as comment on the fact that due to climate change and past wildland management wildfire disturbances are increasing in number and severity – thereby more and more landscapes and the hydrology will be affected by wildfire.

General Comments:

1) Some sort of measurement or estimate in the change of evapotranspiration will really complement and strengthen the conclusions of the manuscript. The work shows how soil moisture evolves differently in burned catchments compared to unburn catchments with infiltration rates or at least surface hydrologic conductivity is largely the same between the catchments and therefore conclude the changes in soil moisture is likely due to changes in evapotranspiration, which is supported by previous literature. However, estimates of changes in evapotranspiration could help provide the 'smoking gun' here. While I acknowledge that accurate ET measurements are hard to come by, changes in NDVI or Modis LAI estimates could really complement this argument perhaps used with a simple ET model.

2) Usually infiltration limited runoff produces fast and flashy hydrographs, whereas groundwater driven stream flow responds more slowly to changes in water table depth. Aside from figure S6 which shows some pictures of flowing streams in the burned catchment and none flowing streams, this discussion is lacking. More about how soil moisture dynamics influences hydrograph shape and duration of flowing catchments could potentially increase the impact of the manuscript. For example, I would expect more a tailing behavior in a hydrograph from a groundwater dominated flow.

Minor comments:

L75-76: Why would the post fire effects particularly influence episodic rainfall and not continual or seasonal precipitation? Especially because it is the change in ET rather than infiltration limited runoff.

L81-82: The specific mechanisms of how soil moisture impacts erosion, debris flows, or landslides is not well defined here.

L252-254: It can be both overland flow and subsurface moisture dynamics contributing to catchment discharge. This isn't really an either-or process.

L304-307: How are plants decreasing local effective hydraulic conductivity? I can see how burned roots could increase infiltration, but this statement seems to say the presence of vegetation actively decreases intrinsic infiltration rates.

L325-338: Some of the material in here calling out specific mechanisms of how soil moisture can control landslides, debris flows, and erosion fits better in the introduction.

L339-342: Seems kind of projection-ish and perhaps a bit out of scope to comment on what the plants will do 5 to 10 years down the road. Also, while this is a longer time scale than storm response it is contradictory to the main lessons learned and your own observations.

L342-344: The 'More research is needed' statement is cliché and not necessary. I suggest omitting, and using this space for something more conclusive and impactful.

Figure 3: It is not easy to discern the trend in the color plots. Is there a way to summarize the soil moisture is a clearer way?

Reviewer #2 (Remarks to the Author):

This is an interesting and very timely study. The authors attempt to use a combination of water isotope data, geophysical surveys and visual observations to compare and contrast two small catchments, one of which suffered wildfire burning in 2020.

The authors claim that they see evidence of enhanced infiltration and shallow groundwater storage in the burned catchment, which contrasts with the state of the unburned catchment. The water isotope data seem to show no difference in behaviour between the two catchments, i.e. the mixing of rapid and slow flow appears to be similar. Visual evidence of higher discharge is claimed in the burned catchment.

GENERAL COMMENTS

The inferred hydrological conceptual model for the two catchments is not clear from the abstract alone. And in fact the abstract's final statement seems to imply that this is intended as a methods paper, which is surely not the case. I think that the authors need to be much clearer in what they are proposing as a hypothesis here.

A major limitation is that (as is often the case in hydrology) this is a $n=1$ experiment. The authors have tried to select two similar catchments to do a paired comparison but we do not know how similar they are prior to the burning. This is compounded by the fact that no data are provided (or available) prior to the wildfire in 2020. As a reviewer it is all too easy to throw in an alternative hypothesis. And so I will (sorry). The subsurface hydrology of these two catchments will be complex given the fractured geology. And we also are forced to assume that the topographic divide marks the catchment boundaries. However, it could be that the two catchments have completely different subsurface structure and hydraulic functioning, irrespective of any burning, i.e. they behaved differently prior to the wildfire simply because of their makeup. And let's say that the topographic divide is not equal to the catchment boundary and so the 'burned catchment' stream receives water from a larger area. Since we have no data to study prior to the wildfire then I struggle to see how we can rule out this alternative interpretation. I feel that the authors need to pay some attention to such alternative hypotheses and try and reject them.

I think that more could be done with the hydraulic conductivity (K_s) data. It is argued (line 129-138) that there is more variation in the burned catchment. I agree with this (visually). And I plotted the additional (Thelma) burned catchment data from table S4 and this also seems to show a similar high variance in K_s to that of Louise (burned), both of which seem to be higher than that of the unburned catchment (Henry) [I ignored the zero values as there will be a finite value of K_s but the authors seem to have assigned a value of zero to those below the detection limit – see specific comment later]. I wondered why the authors had not assessed the variation statistically, and so I decided to do that myself. From an F test I see that the burned (Louise) K_s variance is indeed different from that of the unburned (Henry) K_s at the 5% level. I also did the same with the Thelma (burned) dataset and see that, again, its K_s variance is different to that of the burned catchment at the 5% level. And, reassuringly, there is no difference, at the 5% level, between the variances of the two burned catchments. A simple confirmation of this may be useful. [Note, again, that I discounted the zero values for my simple analysis]. That said, I was a bit unsure

about the interpretation. The authors say that low Ks values in the unburned catchment are probably due to hydrophobic layers due to vegetation. But low values in the burned catchment are due to hydrophobicity from burning. I can understand fire damage leading to hydrophobicity but it seems odd that hydrophobicity is being used to explain behaviour in the two apparently contrasting systems. I think a bit more rigour and robustness in the interpretation of the Ks data is needed.

I like the use of resistivity imaging to illustrate subsurface behaviour and dynamics. However, I wonder how much reliance is on the first (Dec 2020) image. Furthermore, it would be useful to see some attempt at showing changes in resistivity (to infer changes in water content). I realise that the site underwent changes in topography (particularly early 2021) and so I understand why the authors have not carried out a time-lapse inversion. But they could have done this for some periods when the topography was stable. I am assuming, of course, that the same positions were used for electrode locations, which may have not been the case. Either way, I think that attempting to show changes rather than absolute values could be effective. In addition, when carrying out time-lapse surveys we need to be sure that similar coverage is achieved for each survey. Table S2 is useful as it indicates that the size of each dataset (for a specific catchment) is similar. If one survey suffered greater data thresholding/rejection then the coverage will be reduced, in comparison to others. I think that some confirmation that this is not an issue here needs to be made (even if put in the SI).

I wondered why more use wasn't made of the additional burned catchment (Thelma), particularly regarding the resistivity surveys. I got the impression from Figure S8 that these data don't show the same behaviour seen in the main burned catchment (Louise). If so, what does this tell us?

A key interpretation is that "deeper and persistent moisture below the regolith-bedrock interface in the burned catchment may be explained by rapid infiltration of rainwater through preferential flow paths created by burned-out root systems or fractures exposed from the removal of the organic duff layer". Where is this regolith-bedrock interface (i.e. at what depth)? Do we see changes in both burned catchments that are consistent with this statement? I really want to be convinced that there is a difference, notwithstanding my earlier remark that this may not be (conclusively) due to burning.

The streamflow separation is potentially interesting but I'm afraid that I wasn't that convinced by the results. We only see 2 events (Figure 4c). One has no difference between catchments, it is argued that the other suggests a difference but this is based on only a few points. Is this a statistically significant difference? The unburned catchment for the 23 Dec 2021 event shows F_{new} going from a high positive to high negative then back to zero, with only two further points on the graph. How can such erratic behaviour be explained? It seems that this alone is the basis for the statement that the two catchments respond differently. I think that we need to see a more convincing dataset and argument.

SPECIFIC COMMENTS

The abstract needs to be revised to give clear conclusions.

Line 119. State when the Bobcat fire took place. I had to Google it (and see that it was Sept to Dec 2020).

Line 156-160. The visual evidence of difference in streamflow is not that compelling to me (and see comment on photo scale in Figure S6 later). Furthermore, maximum values are stated but there is no explanation (even in the SI) about how these values were obtained.

Line 371. Unsaturated permeability measurements were made with a tension infiltrometer and yet only saturated values are referred to. I wonder why unsaturated values are not used, since they would help confirm macropores.

Line 404. Change "Resipy" to "ResIPy".

Line 405. The reference Binley et al.(2015) doesn't seem appropriate here. I would suggest referring to Binley and Slater, 2020, Resistivity and Induced Polarization: Theory and Applications to the Near-Surface Earth, Cambridge University Press.

Line 405-408. I'm not quite sure what is meant by "Small-scale lateral variability in resistivity was at or below the resolution threshold of our survey, and thus are not distinguishable between potential fractured bedrock outcrops that limit groundwater flow and inversion artifacts from data overfitting". I assume that the authors are trying to explain about signal versus noise in their images but I couldn't quiet work out what was being stated. Some rewording could help here.

Line 437. Missing space before citation.

Figure 2e. Photographs without any scale indicator are of limited value.

Supplementary Information

Line 106 and beyond. The SI refers to "Electrical resistivity tomography" and "ERT" and yet the main text refers to "Electrical resistivity imaging" and EMI. Consistency is needed.

Line 119. Change "Resipy" to "ResIPy".

Line 120. What is meant by "tolerance value between 1 and 1.5"?

Line 131 and beyond. Resistance has units "ohm" not "Ohm".

Line 134. "kitty litter" is an American colloquialism. I understand what it is but some readers may not.

Figure S6. Only one of these photos has a scale reference (someone's foot!) and so they are difficult to compare.

Table S4. What does a zero hydraulic conductivity mean? I suspect that it is just a value below the detection limit. This needs to be stated along with the detection limit or measurement threshold.

SUMMARY

As I said right at the start, this is a very timely study and it could attract attention from a wide readership. However, in order to be ready for publication I believe that a more rigorous analysis and convincing argument for a specific hypothesis should be made. I suspect that the information lies within the data but at the moment the case is not made.

Andrew Binley
1-Sept-2022

Reviewer #3 (Remarks to the Author):

General Comments

This is an interesting paper that presents findings from a survey of soil resistivity and O isotopes in burned and unburned watersheds of Southern California. The authors consider factors that contribute to an apparent conundrum where post-fire increases in surface runoff occur simultaneously with increases in groundwater recharge. The mechanisms they credit for the increase in subsurface water inputs include rapid infiltration along burned root macropores and fractured bedrock during storm events. The work has important implications for post-fire hydrology that may impact debris flow initiation and post-fire erosion, which can be significant in

the dry ravel-dominated bedrock landscape of the study region.

The authors compared patterns in resistivity and isotope chemistry of a burned and an unburned catchment over the course of 2 years. Some of the graphic patterns of post-fire soil resistivity suggest catchment scale differences in subsurface wetting and drying though the generalizability of this information is constrained by the lack of pre-fire data. It is impossible to rule out the possibility that pre-fire differences in infiltration, bedrock depth, fracture, subsurface storage, rather than post-fire changes underlie the observations.

The author's contention that storm water can rapidly move into the subsurface via fractured bedrock and burned root macropores is certainly plausible. Fire is known to increase rock weathering in some systems areas, and it creates dead roots. However, both fractured rock and macropores created by dead roots would both be present in both burned and unburned areas. In these fire-prone ecosystems, with relatively short wildfire return interval, burned root macropores should be ubiquitous.

Without information about the density of these features in both burned and unburned areas, it is not possible to evaluate their potential significantly alter post-fire infiltration or their capacity to capture water during rainfall events. Estimates of infiltration rates along root channels and through fragmented rock combined with information on the density of these features would strengthen the authors interpretations of the differences in subsurface resistivity between the burned and unburned catchments. Such estimates would also help readers interpret to what extent these findings may be relevant to other geologic or geomorphic settings.

The current manuscript presents an intriguing case study, but it is my opinion that additional information is required to present a convincing argument that the physical features credited with rapid subsurface flow have the capacity to move sufficient water to create the observed patterns.

Specific Comments

Page 3 Few of these hypothesized factors are actually measured in the study.

Page 4 Results 'spatially variable hydrophobic layer from fire.' Please clarify meaning. Do you suggest that fire made the pre-fire layer more variable or that fire created a variable layer or??

Based on the previous statement about hydrophobicity being linked to vegetation and duff, does the statement indicate that removal of vegetation translates directly to the loss of hydrophobicity. What process is responsible: direct combustion of the hydrophobic compounds or leaching of them due to loss of canopy interception and ET, or ? Please clarify.

SI Page 4 - What about salinity or electrical conductance and how it may differ among sites or across depths?

RESPONSE TO REVIEWER COMMENTS

Reviewer #1 (Remarks to the Author):

This manuscript evaluates subsurface moisture dynamics following fire in a chaparral system and compares a severely burned catchment with a nearby unburned catchment using electrical resistivity imaging and stable isotopic analysis. Much of the previously published research on fire effects to hydrology has been focused on surface runoff generated from reduced infiltration rates often attributed to hydrophobic soils and until recently relatively few papers have considered subsurface effects from wildland fire. This paper does an admirable job conceptualizing how fire affects will change subsurface moisture conditions, which in turn is reflected in catchment discharge magnitude and duration. This conceptualization is supported in this manuscript using paired catchments and detailed resistivity and isotopic observations. The manuscript is well written, and I find the methods used to be robust. It is my opinion that this paper deserves publication as it adds to the growing research that describes coupled surface and subsurface hydrology considering fire disturbance. As a result, most of my comments are editorial in nature with the aim to improve the paper. I will say that publishing in Nature Communication deserves a level of scientific impact and novelty that needs to be evident. Here the authors could improve this aspect by identifying which processes and concepts will apply or add to knowledge outside of the Chaparral system as well as comment on the fact that due to climate change and past wildland management wildfire disturbances are increasing in number and severity – thereby more and more landscapes and the hydrology will be affected by wildfire.

We thank the reviewer for their constructive comments and appreciate the suggestion to emphasize more clearly the wide relevance of this work because of changing fire regimes. As the reviewer points out, considering subsurface hydrology in the context of increasing wildfire frequency is important to understand cascading post-fire hazards such as shallow landslide and flooding, as well as the recovery interval for ecosystems. In the Introduction, we note that wildfire frequency is expected to increase in the future, which emphasizes the importance of our study:

“Wildfire frequency and size are expected to increase as global climate change affects seasonal temperature and precipitation intensity extremes (Donat et al., 2016; Mann et al., 2016; Westerling et al., 2006; Westerling & Bryant, 2008). More frequent and intense fires could exacerbate floods and debris flows, increase erosion, and imperil water resources (Dennison et al., 2014; Flannigan et al., 2009; Hallema et al., 2018). “

In revision, we have changed the concluding sentence of the manuscript discussion linking back to this idea:

“As wildfire frequency increases in Southern California and other locations, it will become increasingly important to consider subsurface hydrology in the context of post-fire cascading hazards (e.g., shallow landslides, flooding) and ecosystem recovery.”

We also appreciate the reviewer's comments on broadening the implications of this study beyond chaparral ecosystems. It is difficult to extrapolate our findings to other settings because our study area and the associated geomorphological and hydrological processes may be specific to chaparral environments such as southern California. However, in offering an alternative conceptual model (Figure 2) to the broadly accepted paradigm of infiltration-excess overland flow driving post-fire hazards in southern California, we think our paper should stimulate future post-fire studies to further consider subsurface hydrology more carefully in other locations.

General Comments:

1) Some sort of measurement or estimate in the change of evapotranspiration will really complement and strengthen the conclusions of the manuscript. The work shows how soil moisture evolves differently in burned catchments compared to unburn catchments with infiltration rates or at least surface hydrologic conductivity is largely the same between the catchments and therefore conclude the changes in soil moisture is likely due to changes in evapotranspiration, which is supported by previous literature. However, estimates of changes in evapotranspiration could help provide the 'smoking gun' here. While I acknowledge that accurate ET measurements are hard to come by, changes in NDVI or Modis LAI estimates could really complement this argument perhaps used with a simple ET model.

We thank the reviewer for this suggestion. We have now added a section on evapotranspiration using ET data from NASA's ECOSTRESS satellite, which began operation in 2018 and provides high temporal resolution ET data. As the reviewer notes we should expect, the new results show very similar evapotranspiration trends prior to the fire in the catchments, with significant changes postfire. ET appears to rebound in the burned catchments in the spring of WY2, which corresponds to increased drying from ERT data during that same time period.

2) Usually infiltration limited runoff produces fast and flashy hydrographs, whereas groundwater driven stream flow responds more slowly to changes in water table depth. Aside from figure S6 which shows some pictures of flowing streams in the burned catchment and none flowing streams, this discussion is lacking. More about how soil moisture dynamics influences hydrograph shape and duration of flowing catchments could potentially increase the impact of the manuscript. For example, I would expect more a tailing behavior in a hydrograph from a groundwater dominated flow.

Unfortunately, due to field conditions we were unable to measure discharge which would have allowed for a quantitative hydrograph separation and, as the reviewer suggests, for characterizing the tailing of the hydrograph. However, as noted in figure 4b, we saw a very long tailed hydrograph (based on field observations of running water) in WY2 after multiple storms in December. This has also been discussed in the original and revised manuscript further in the Discussion section, highlighting how the longer flow duration may relate to increased groundwater availability:

“We find increased subsurface storage immediately following a single storm as well as prolonged streamflow after significant rainfall (Figure 4b), consistent with observations of long-term hydrological effects of wildfire that include increased baseflow and aquifer recharge.”

Minor comments:

L75-76: Why would the post fire effects particularly influence episodic rainfall and not continual or seasonal precipitation? Especially because it is the change in ET rather than infiltration limited runoff.

We agree with the reviewer and have removed this sentence from the manuscript.

L81-82: The specific mechanisms of how soil moisture impacts erosion, debris flows, or landslides is not well defined here.

There is a rich body of literature that addresses the link between soil moisture and debris flows and landslides, particularly in Southern California and post-wildfire. Given that we did not directly observe or study debris flows, we chose not to elaborate in the Introduction beyond citation to the relevant prior work.

L252-254: It can be both overland flow and subsurface moisture dynamics contributing to catchment discharge. This isn't really an either-or process.

We agree and have clarified the text to reflect this point in several locations. The following text in our revised Discussion reconciles how surface and subsurface flow are simultaneously contributing to streamflow:

“Discharge observations (Figure S2, Figure S3) and water isotope data (Figure 4) indicate that this subsurface water is dynamic and contributes to surface flow, which challenges the longstanding paradigm that in Southern California, post-fire storm discharge originates only from infiltration-excess overland flow.”

“Rainfall intensities exceeded minimum K_{fs} values in both catchments during storms that generated streamflow (Figure 2d; Figure S4), suggesting that infiltration-excess overland flow was contributing to streamflow in both catchments (see further discussion in the Supplement). Yet our data suggest that a substantial portion of precipitation also infiltrated into the subsurface and, importantly, that some of this stored water was then mobilized back as streamflow during storms (Figure 2c).”

L304-307: How are plants decreasing local effective hydraulic conductivity? I can see how burned roots could increase infiltration, but this statement seems to say the presence of vegetation actively decreases intrinsic infiltration rates.

The reviewer makes a good point, and to avoid confusion we removed this statement about hydraulic conductivity.

L325-338: Some of the material in here calling out specific mechanisms of how soil moisture can control landslides, debris flows, and erosion fits better in the introduction.

We were strictly trying to show the implications of subsurface reservoirs, and since we did not study their actual effects on erosion and landslides we did not feel that it was appropriate to add text on this topic to the Introduction. Rather we felt that future work in these areas could be stimulated by our findings, thus making the Discussion section a better fit.

L339-342: Seems kind of projection-ish and perhaps a bit out of scope to comment on what the plants will do 5 to 10 years down the road. Also, while this is a longer time scale than storm response it is contradictory to the main lessons learned and your own observations.

We thank the reviewer for this comment. The section of original text the reviewer is referring to is the following:

“Shallow water storage may also facilitate rapid vegetation regrowth, affecting post-fire ecology and potentially reducing groundwater recharge through increased evapotranspiration and stabilizing hillslopes on these same timescales (Obrist et al., 2004; Silberstein et al., 2013).”

We disagree with the reviewer that this is “projection-ish” as we saw revegetation over the course of WY2, as noted in the Discussion section:

“In the burned catchments in our study, water persisted at the near surface between WYs, but towards the end of WY2, drying was co-located with observed vegetation regrowth.”

Revegetation in the burned catchments is now also reflected in the new ET data, which show a rebound in ET in WY2, a point we have now added to the Discussion in our revision:

“In WY2, ET rates rebounded in the burned catchments, indicating substantial increase in vegetation growth that is consistent with near-surface changes in resistivity in WY2 (Figure 3n).”

That being said, we agree that aspects of our previous wording was confusing and have reworded our the aspects that seemed contradictory, as follows:

“Shallow water storage in our study site appears to facilitate rapid vegetation regrowth, affecting post-fire ecology and potentially stabilizing hillslopes in the process (Obrist et al., 2004; Silberstein et al., 2013).”

L342-344: The ‘More research is needed’ statement is cliché and not necessary. I suggest omitting, and using this space for something more conclusive and impactful.

We have removed this comment from our text and added a stronger concluding statement.

Figure 3: It is not easy to discern the trend in the color plots. Is there a way to summarize the soil moisture is a clearer way?

We thank the reviewer for this comment, which is similar to a comment made by Reviewer 2. We have changed the figure to show % change in resistivity from the initial survey (Dec 2020) to better show trends in the changing resistivity.

Reviewer #2 (Remarks to the Author):

This is an interesting and very timely study. The authors attempt to use a combination of water isotope data, geophysical surveys and visual observations to compare and contrast two small catchments, one of which suffered wildfire burning in 2020.

We are glad that the reviewer recognizes the timeliness and novelty of our work.

The authors claim that they see evidence of enhanced infiltration and shallow groundwater storage in the burned catchment, which contrasts with the state of the unburned catchment. The water isotope data seem to show no difference in behaviour between the two catchments, i.e. the mixing of rapid and slow flow appears to be similar. Visual evidence of higher discharge is claimed in the burned catchment.

GENERAL COMMENTS

The inferred hydrological conceptual model for the two catchments is not clear from the abstract alone. And in fact the abstract's final statement seems to imply that this is intended as a methods paper, which is surely not the case. I think that the authors need to be much clearer in what they are proposing as a hypothesis here.

We have re-phrased the abstract to state our hypothesis and findings more clearly. We thank the reviewer for pointing out that the prior phrasing had suggested that this is a methods paper, which was not our intent! We hope the revisions address this concern.

A major limitation is that (as is often the case in hydrology) this is a $n=1$ experiment. The authors have tried to select two similar catchments to do a paired comparison but we do not know how similar they are prior to the burning. This is compounded by the fact that no data are provided (or available) prior to the wildfire in 2020. As a reviewer it is all too easy to throw in an alternative hypothesis. And so I will (sorry). The subsurface hydrology of these two catchments will be complex given the fractured geology. And we also are forced to assume that the topographic divide marks the catchment boundaries. However, it could be that the two catchments have completely different subsurface structure and hydraulic functioning, irrespective of any burning, i.e. they behaved differently prior to the wildfire simply because of their makeup. And let's say that the topographic divide is not equal to the catchment boundary and so the 'burned catchment' stream receives water from a larger area. Since we have no data to study prior to the wildfire then I struggle to see how we can rule out this alternative interpretation. I feel that the authors need to pay some attention to such alternative hypotheses and try and reject them.

Paired catchments are a common tool in hydrology and are often used to understand perturbations including post-mining, logging and wildfire disturbances, with a history extending back over 100 years (a point out we now make in the Introduction of the revised submission). While certainly imperfect, this framework well-established, widely-vetted, and generally-

accepted. In the case of our study, we lack pre-fire hydrologic monitoring data as the reviewer notes, but we have several other lines of evidence to establish the comparability of our study catchments. Moreover, we have in fact studied three, not just two, catchments. In our initial submission, data from the third catchment was left in the Supplement in part because we were unable to continue studying this site in the second water year (due to field hand availability). In response to the Reviewer's comment, we have now added the 3rd catchment back into the ERI discussion to support our findings that the burned catchments differed from the unburned catchment because of fire history. As shown in SI Figure 1 and Table 1, we have included relevant pre-fire data that could result in a different hydrologic response among the three catchments. All three catchments (Thelma, Louise and Henry) have similar slope distributions, pre-fire evapotranspiration rates, and drainage areas, and all three are mapped within the same geologic units. Previous work by Neeley and DiBiase (2020) show broadly similar bedrock fracture density across the SGM, that the gneiss in this region is pervasively sheared and fractured, and that large differences in fracturing over such a small distance between these catchments is not expected. Moreover, Louise and Henry are located on the same side of the drainage divide. We consider the topographic divide to be the hydrologic divide between catchments in terms of surface water input. The third catchment (Thelma) has different aspect, vegetation and topographic location with respect to the divide compared to the other two catchments, and these factors might be expected to produce a different hydrologic response to storms if the Reviewer's arguments are correct. However, the post-fire response in Thelma is similar to the response in Louise, which was also burned. Given that we see a similar response in both burned catchments, we argue that the Bobcat Fire is dominantly responsible for changes in hydrology when comparing Louise, Thelma and Henry.

Citation: Neely, A. B., & DiBiase, R. A. (2020). Drainage area, bedrock fracture spacing, and weathering controls on landscape-scale patterns in surface sediment grain size. *Journal of Geophysical Research: Earth Surface*, 125(10), e2020JF005560.

I think that more could be done with the hydraulic conductivity (K_s) data. It is argued (line 129-138) that there is more variation in the burned catchment. I agree with this (visually). And I plotted the additional (Thelma) burned catchment data from table S4 and this also seems to show a similar high variance in K_s to that of Louise (burned), both of which seem to be higher than that of the unburned catchment (Henry) [I ignored the zero values as there will be a finite value of K_s but the authors seem to have assigned a value of zero to those below the detection limit – see specific comment later]. I wondered why the authors had not assessed the variation statistically, and so I decided to do that myself. From an F test I see that the burned (Louise) K_s variance is indeed different from that of the unburned (Henry) K_s at the 5% level. I also did the same with the Thelma (burned) dataset and see that, again, its K_s variance is different to that of the burned catchment at the 5% level. And, reassuringly, there is no difference, at the 5% level, between the variances of the two burned catchments. A simple confirmation of this may be useful. [Note, again, that I discounted the zero values for my simple analysis]. That said, I was a bit unsure about the interpretation. The authors say that low K_s values in the unburned catchment are probably due to hydrophobic layers due to vegetation. But low values in the burned catchment are due to hydrophobicity from burning. I can understand fire damage leading

to hydrophobicity but it seems odd that hydrophobicity is being used to explain behaviour in the two apparently contrasting systems. I think a bit more rigour and robustness in the interpretation of the Ks data is needed.

We thank the reviewer for this suggestion; our USGS internal review included a similar suggestion. In revision, we added an assessment of the statistical variance between the field saturated hydraulic conductivity data in all three catchments (now included in Figure 2) using an F-test. Using an alpha value of 0.05, we find no statistical difference between Kfs values in Thelma and Louise, and significant difference between both Thelma and Henry and Louise and Henry, which is in line with the Reviewer's analysis. These values are shown here and reported in the revised manuscript in Supplementary Table 5:

Variances: Louise: 871.72, Thelma: 688.30, Henry: 201.96
Louise and Henry: F-score: 4.29, P-value: 0.0026
Thelma and Henry: F-score: 3.36, P-value: 0.0083
Louise and Thelma: F-score 1.28, P-value:0.2937
P-value of <5% (0.05) indicates significantly different populations

Note that, in plotting the data, we took the log of our Kfs data for better data visualization and given the large variability in the burned catchment values. However, for the F-test of statistical difference, we did not log-transform the data, though we did remove outliers beyond three standard deviations.

Finally, with regards to the reviewer's specific comment on 0 Kfs values: if no water infiltrated after 45 minutes, a value of zero was assigned, following methods by Wall et al. (2020) - as stated below we have added this specification to the revised text.

Citation: Wall, S. A., Roering, J. J., & Rengers, F. K. (2020). Runoff-initiated post-fire debris flow Western Cascades, Oregon. *Landslides*, 17(7), 1649-1661.

I like the use of resistivity imaging to illustrate subsurface behaviour and dynamics. However, I wonder how much reliance is on the first (Dec 2020) image. Furthermore, it would be useful to see some attempt at showing changes in resistivity (to infer changes in water content). I realise that the site underwent changes in topography (particularly early 2021) and so I understand why the authors have not carried out a time-lapse inversion. But they could have done this for some periods when the topography was stable. I am assuming, of course, that the same positions were used for electrode locations, which may have not been the case. Either way, I think that attempting to show changes rather than absolute values could be effective. In addition, when carrying out time-lapse surveys we need to be sure that similar coverage is achieved for each survey. Table S2 is useful as it indicates that the size of each dataset (for a specific catchment) is similar. If one survey suffered greater data thresholding/rejection then the coverage will be reduced, in comparison to others. I think that some confirmation that this is not an issue here needs to be made (even if put in the SI).

The reviewer makes a good point, which is similar to comments from Reviewer #1. We have changed the figure to show the time lapse values using December 2020 as the baseline survey. The individual surveys are now included in the Supplement in Figures S8 and S9. We also report the % of data kept after rejecting anomalous values in Table S2, which is similar between surveys and is not expected to result in a difference in coverage.

I wondered why more use wasn't made of the additional burned catchment (Thelma), particularly regarding the resistivity surveys. I got the impression from Figure S8 that these data don't show the same behaviour seen in the main burned catchment (Louise). If so, what does this tell us?

On reflection, we agree with the reviewer and have added data from the third catchment to the resistivity discussion. We originally elected to exclude this catchment because we did not sample for rain and streamflow due to limited field assistance and challenging logistics of accessing the site, and for the same reason, we did not measure resistivity at this site as frequently during WY2. After taking timelapse inversions, we see similar behavior in both burned catchments, which we agree strengthens our manuscript (especially in light of the reviewer's concern about other potential factors driving differences between catchments as discussed above).

A key interpretation is that "deeper and persistent moisture below the regolith-bedrock interface in the burned catchment may be explained by rapid infiltration of rainwater through preferential flow paths created by burned-out root systems or fractures exposed from the removal of the organic duff layer". Where is this regolith-bedrock interface (i.e. at what depth)? Do we see changes in both burned catchments that are consistent with this statement? I really want to be convinced that there is a difference, notwithstanding my earlier remark that this may not be (conclusively) due to burning.

The regolith-bedrock interface was measured in the field using a measuring rod. The depth at which we could no longer penetrate the underlying bedrock was measured as the regolith-bedrock interface. After reflecting on the reviewer's comment as well as other reviewers' comments, we have decided to remove mention of this boundary from the paper, as our point measurements were scarce along the ERI profiles with broad interpolations between points, and changes in weathering and fracture density are likely more variable both laterally and vertically than what we have measured. Removing this minor component of our work does not affect our interpretations or conclusions.

The streamflow separation is potentially interesting but I'm afraid that I wasn't that convinced by the results. We only see 2 events (Figure 4c). One has no difference between catchments, it is argued that the other suggests a difference but this is based on only a few points. Is this a statistically significant difference? The unburned catchment for the 23 Dec 2021 event shows Fnew going from a high positive to high negative then back to zero, with only two further points on the graph. How can such erratic behaviour be explained? It seems that this alone is the basis

for the statement that the two catchments respond differently. I think that we need to see a more convincing dataset and argument.

We appreciate this comment, which seems to stem from a misunderstanding of the interpretation. We reported water isotope values from 7 storm events (not 2, as the reviewer implies), and our primary conclusion is that there is no difference in isotope values between the catchments (despite there being a difference in discharge), which led to our conclusion that more water from the subsurface/previous storms was entering the burned catchment. We are only able to use a quantitative mixing model (Fnew) for two events due to data availability (because of intermittent rain and streamflow), but the rest of our isotopic data show a similar pattern to what we see in those events. In our revision, we have worked to clarify the wording of our text to avoid potential misunderstanding in the future, including the following revised statements:

“However, during individual storms, we found that streamflow $\delta^{18}\text{O}$ values could differ from contemporaneous precipitation $\delta^{18}\text{O}$ values (Figure 5). For example, on 10 March 2021, 25 October 2021, 14 December 2021 and 28 March 2022, the streamflow and precipitation samples had similar $\delta^{18}\text{O}$ distributions, while the 15 March 2021, 23 December 2021 streamflow samples, as well as a streamflow sample collected after the 28 December 2021 storm, had offset distributions from rainfall (Figure 5).”

“Isotopic similarities in streamflow between the unburned catchment and the sampled burned catchment in our study show that the same proportions of stored water and rainfall are driving streamflow in the two catchments (Figures 4a, 5)”

“Stream and precipitation samples from 10 March 2021, 14 December 2021 and 28 March 2022 are similar, indicating that runoff during that event was likely generated in large part from overland flow. 15 March 2021 stream samples in both catchments and 23-24 December 2021 stream samples in the unburned catchment show a distinct signature from cotemporal precipitation”

SPECIFIC COMMENTS

The abstract needs to be revised to give clear conclusions.

We thank the reviewer for this suggestion and have revised the wording of the Abstract with this point in mind, as follows:

“Electric resistivity imaging shows that in the burned catchments after the fire, rainfall infiltrated deep into the weathered bedrock and persisted. Isotope data from stormflow suggests that the amount of mixing of surface and subsurface water during storm events was similar in burned and unburned catchments, despite higher discharge following burning. Therefore, both surface runoff and infiltration likely increased in tandem”

Line 119. State when the Bobcat fire took place. I had to Google it (and see that it was Sept to Dec 2020).

We made this change as suggested.

Line 156-160. The visual evidence of difference in streamflow is not that compelling to me (and see comment on photo scale in Figure S6 later). Furthermore, maximum values are stated but there is no explanation (even in the SI) about how these values were obtained.

The reviewer seems to have missed the description of this methodology in the Methods section [Lines 416-427] and visualization in Figure S7. Further methodology is explained in the Methods section.

Line 371. Unsaturated permeability measurements were made with a tension infiltrometer and yet only saturated values are referred to. I wonder why unsaturated values are not used, since they would help confirm macropores.

In post-fire literature, measurements from tension infiltrometer have been referred to as “field-saturated hydraulic conductivity”. In keeping with that norm, we refer to it as such, although this was misstated in the Methods section (as noted in the reviewer’s comment). We have changed the text in the Methods to refer to field saturated hydraulic conductivity to be consistent throughout.

Line 404. Change “Resipy” to “ResIPy”.

We made this change as suggested.

Line 405. The reference Binley et al.(2015) doesn’t seem appropriate here. I would suggest referring to Binley and Slater, 2020, Resistivity and Induced Polarization: Theory and Applications to the Near-Surface Earth, Cambridge University Press.

Fixed.

Line 405-408. I’m not quite sure what is meant by “Small-scale lateral variability in resistivity was at or below the resolution threshold of our survey, and thus are not distinguishable between potential fractured bedrock outcrops that limit groundwater flow and inversion artifacts from data overfitting”. I assume that the authors are trying to explain about signal versus noise in their images but I couldn’t quiet work out what was being stated. Some rewording could help here.

The reviewer has correctly interpreted our meaning. We have reworded the text for clarity.

Line 437. Missing space before citation.

Fixed.

Figure 2e. Photographs without any scale indicator are of limited value.

We added a scale bar from field measurements.

Supplementary Information

Line 106 and beyond. The SI refers to “Electrical resistivity tomography” and “ERT” and yet the main text refers to “Electrical resistivity imaging” and EMI. Consistency is needed.

We thank the reviewer for pointing out this inconsistency. We now consistently use “electrical resistivity imaging” and ERI.

Line 119. Change “Resipy” to “ResIPy”.

Changed.

Line 120. What is meant by “tolerance value between 1 and 1.5”?

This is the allowable error (RMSE) between the data and the model.

Line 131 and beyond. Resistance has units “ohm” not “Ohm”.

Changed.

Line 134. “kitty litter” is an American colloquialism. I understand what it is but some readers may not.

We changed the phrasing to “cat litter” which is hopefully clearer.

Figure S6. Only one of these photos has a scale reference (someone’s foot!) and so they are difficult to compare.

We have added scales based on field measurements.

Table S4. What does a zero hydraulic conductivity mean? I suspect that it is just a value below the detection limit. This needs to be stated along with the detection limit or measurement threshold.

Hydraulic conductivity of zero means no infiltration occurred in 45 minutes. The text has been revised to reflect this.

SUMMARY

As I said right at the start, this is a very timely study and it could attract attention from a wide readership. However, in order to be ready for publication I believe that a more rigorous analysis and convincing argument for a specific hypothesis should be made. I suspect that the information lies within the data but at the moment the case is not made.

We have worked hard to address the reviewer’s comments and think the revised manuscript is stronger as a result. We appreciate their thoughtful, constructive, and detailed review.

Andrew Binley

1-Sept-2022

Reviewer #3 (Remarks to the Author):

General Comments

This is an interesting paper that presents findings from a survey of soil resistivity and O isotopes in burned and unburned watersheds of Southern California. The authors consider factors that contribute to an apparent conundrum where post-fire increases in surface runoff occur simultaneously with increases in groundwater recharge. The mechanisms they credit for the increase in subsurface water inputs include rapid infiltration along burned root macropores and fractured bedrock during storm events. The work has important implications for post-fire hydrology that may impact debris flow initiation and post-fire erosion, which can be significant in the dry ravel-dominated bedrock landscape of the study region.

The authors compared patterns in resistivity and isotope chemistry of a burned and an unburned catchment over the course of 2 years. Some of the graphic patterns of post-fire soil resistivity suggest catchment scale differences in subsurface wetting and drying though the generalizability of this information is constrained by the lack of pre-fire data. It is impossible to rule out the possibility that pre-fire differences in infiltration, bedrock depth, fracture, subsurface storage, rather than post-fire changes underlie the observations.

Reviewer #2 raised a similar question. We have evaluated potential sources of inter-catchment variability that could contribute to different hydrological response including slope, aspect, pre-fire evapotranspiration rates, normalized channel steepness and lithology (as shown in SI Figure 1 and Table S1). In revision, we have added discussion of the second burned catchment (Thelma). All three catchments have similar slope distributions, pre-fire evapotranspiration rates, and drainage areas, and all three are mapped within the same geologic units. Louise and Henry are located on the same side of the primary east-west drainage divide. The third catchment (Thelma) has different physical characteristics (aspect, vegetation and topographic location with respect to the divide) compared to the other two catchments that might be expected to produce a different hydrologic response to storms; however, the response in Thelma is similar to the response in Louise, which was also burned. Prior work by Neeley and DiBiase (2020) and DiBiase et al. (2018) indicate that fractures are pervasive in the highly sheared gneisses of the SGM and large changes in fracturing are not expected over such a small region. Given that we see a similar response to storms in both burned catchments, we argue that the Bobcat Fire is dominantly responsible for changes in hydrology when comparing the three study catchments.

Citations:

Neely, A. B., & DiBiase, R. A. (2020). Drainage area, bedrock fracture spacing, and weathering controls on landscape-scale patterns in surface sediment grain size. *Journal of Geophysical Research: Earth Surface*, 125(10), e2020JF005560.

DiBiase, R. A., Rossi, M. W., & Neely, A. B. (2018). Fracture density and grain size controls on the relief structure of bedrock landscapes. *Geology*, 46(5), 399-402.

The author's contention that storm water can rapidly move into the subsurface via fractured bedrock and burned root macropores is certainly plausible. Fire is known to increase rock

weathering in some systems areas, and it creates dead roots. However, both fractured rock and macropores created by dead roots would both be present in both burned and unburned areas. In these fire-prone ecosystems, with relatively short wildfire return interval, burned root macropores should be ubiquitous.

Without information about the density of these features in both burned and unburned areas, it is not possible to evaluate their potential significantly alter post-fire infiltration or their capacity to capture water during rainfall events. Estimates of infiltration rates along root channels and through fragmented rock combined with information on the density of these features would strengthen the authors interpretations of the differences in subsurface resistivity between the burned and unburned catchments. Such estimates would also help readers interpret to what extent these findings may be relevant to other geologic or geomorphic settings.

The current manuscript presents an intriguing case study, but it is my opinion that additional information is required to present a convincing argument that the physical features credited with rapid subsurface flow have the capacity to move sufficient water to create the observed patterns.

We agree with the reviewer's assessment that data quantifying fracture densities and associated infiltration rates in the burned and unburned catchments would be interesting and would help further develop the understanding of post-fire hydrology in this setting. However, the field effort required to generate robust datasets of these parameters would be enormous and would likely require new methods development. We view this endeavor as being well beyond the scope of the study, and in any case any such data collected now at our field sites would likely have limited utility as three years have now elapsed since the fire with significant revegetation recovery ongoing. We hope that our work may contribute to further such efforts in the future.

Although we do not have detailed field data on these parameters, we are not the first work to invoke specific mechanisms of preferential flow in burned catchments. Preferential flow via macropores has been modelled and invoked by numerous prior studies. For example, Nyman et al. 2010 and Nyman et al. 2014 model the role of various infiltration mechanisms and find macropores in burned catchments as important and rapid contributors to subsurface flow. Doerr et al.'s 2000 review paper invokes macropores through root channels and fractures as potential flow paths beyond the soil matrix citing Garkaklis et al. 1998, Ferreira et al. 2000, and Burch et al. 1989. Shakesby and Doerr 2006 also cite macropore or root characteristics as important changes to subsurface hydrology. Leslie et al. 2014 found that while the creation of soil pipes occurs generally during the dying and decay of roots, these become connected to the surface via fire conditions, then acting as macropores for water transport. Cardenas and Kanarek 2014 also invoke macropores in dead or dying roots to explain why they see increased subsurface moisture postfire and indicate that while live roots might also have this effect, they are much more likely to consume moisture - preventing it moving deeper into the subsurface, which appears to happen in our unburned catchment (Figure 3). Stoof et al. 2014 showed that in burned vs unburned plots in Portugal, preferential flow through finger flow occurs in burned

catchments more often, likely due to the dryness of the soils. Doerr et al. 2000 also cite finger flow in soil matrices. We have included more about the potential for finger flow as a mechanism in the paper.

In summary, we feel that this extensive prior literature provides a solid foundation for identifying plausible mechanisms to explain our observations. While further quantifying the rate of infiltration associated with each of these different mechanisms would be an interesting future study, we see it as being beyond the scope of our work which we argue has identified how important infiltration is in post-fire settings. We have clarified these points and added further references from the list below into the Introduction.

Citations

Burch, G.J., Moore, I.D., Burns, J., 1989. Soil hydrophobic effects on infiltration and catchment runoff. *Hydrological Processes* 3, 211–222.

Cardenas, M. Bayani, and Michael R. Kanarek. "Soil moisture variation and dynamics across a wildfire burn boundary in a loblolly pine (*Pinus taeda*) forest." *Journal of hydrology* 519 (2014): 490-502.

Doerr, Stefan H., R. A. Shakesby, and R.P.D. Walsh. "Soil water repellency: its causes, characteristics and hydro-geomorphological significance." *Earth-Science Reviews* 51, no. 1-4 (2000): 33-65.

Ferreira, A.J.D., Coelho, C.O.A., Walsh, R.P.D., Shakesby, R.A., Ceballos, A., Doerr, S.H., 2000. Hydrological implications of soil water repellency in *Eucalyptus globulus* and *Pinus pinaster* forests, north-central Portugal. *Journal of Hydrology*,

Garkaklis, M.J., Bradley, J.S., Wooler, R.D., 1998. The effects of Woylie (*Bettongia penicillata*) foraging on soil water repellency and water infiltration in heavy textured soils in southwestern Australia. *Australian Journal of Ecology* 23, 492–496

Leslie, Ian N., Robert Heinse, Alistair MS Smith, and Paul A. McDaniel. "Root decay and fire affect soil pipe formation and morphology in forested hillslopes with restrictive horizons." *Soil Science Society of America Journal* 78, no. 4 (2014): 1448-1457.

Nyman, Petter, Gary J. Sheridan, Hugh G. Smith, and Patrick NJ Lane. "Modeling the effects of surface storage, macropore flow and water repellency on infiltration after wildfire." *Journal of Hydrology* 513 (2014): 301-313.

Nyman, Petter, Gary Sheridan, and Patrick NJ Lane. "Synergistic effects of water repellency and macropore flow on the hydraulic conductivity of a burned forest soil, south-east Australia." *Hydrological Processes* 24, no. 20 (2010): 2871-2887.

Shakesby, Richard A., and Stefan H. Doerr. "Wildfire as a hydrological and geomorphological agent." *Earth-Science Reviews* 74, no. 3-4 (2006): 269-307.

Stoof, C. R., E. C. Slingerland, W. Mol, J. Van Den Berg, P. J. Vermeulen, A. J. D. Ferreira, C. J. Ritsema, J.-Y. Parlange, and T. S. Steenhuis. "Preferential flow as a potential mechanism for fire-induced increase in streamflow." *Water Resources Research* 50, no. 2 (2014): 1840-1845.

Specific Comments

Page 3 Few of these hypothesized factors are actually measured in the study.

We infer that the reviewer is referring to following statement:

"We hypothesize that the combined effects of hydrophobic soils, decreased surface roughness, rapid rainwater infiltration through macropores, and reduced evapotranspiration due to vegetation loss may lead to both increased surface runoff and groundwater recharge, with potential dynamic connections between these reservoirs."

The reviewer brings up a valid point. We have removed this statement from the revised manuscript and instead now posit well-studied mechanisms in the Discussion that explain our findings, with greater emphasis on citing prior literature (also see our reply above). We have also included new analysis on evapotranspiration. This paper is meant to challenge the conceptual paradigm that post-fire discharge is primarily due to infiltration-excess surface runoff in southern California. Measuring the individual effects of hydrophobic soils, surface roughness, and infiltration through macropores was beyond the scope of this study, but we encourage future studies to pursue an in-depth look at these mechanisms for post-fire hydrologic change.

Our revised Discussion related to this point now reads as follows:

"In comparison, deeper and persistent water the burned catchments may be explained by rapid infiltration of rainwater through flow paths following preferential finger flow or created by burned-out root systems or fractures exposed from the removal of the organic duff layer (Nyman et al., 2010; Stoof et al., 2014; Cardenas and Kanarek, 2014) (Figure 2e), as well as significant loss of evapotranspiration during WY1 (Silva et al., 2006; Stoof et al., 2012)."

Page 4 Results 'spatially variable hydrophobic layer from fire.' Please clarify meaning. Do you suggest that fire made the pre-fire layer more variable or that fire created a variable layer or??

Based on the previous statement about hydrophobicity being linked to vegetation and duff, does the statement indicate that removal of vegetation translates directly to the loss of hydrophobicity. What process is responsible: direct combustion of the hydrophobic compounds or leaching of them due to loss of canopy interception and ET, or ? Please clarify.

We understand why the reviewer found this sentence confusing. In the process of completing revisions in response to other reviewers, we have completed a statistical test of variability in unsaturated hydraulic conductivity (K_{fs}) values measured between catchments, which revealed substantially more spatial variability in K_{fs} in the burned catchment than the unburned catchment. We have changed the Results to read the following:

“The burned catchments had more variable K_{fs} (significant at the 5% level based on an F-test; Table S5) than the unburned catchment, indicating substantial spatial variability in potential infiltration-excess surface runoff and infiltration.”

SI Page 4 - What about salinity or electrical conductance and how it may differ among sites or across depths?

We have addressed the possibility of salinity affecting our resistivity data with respect to interpreting negative changes in resistivity as the addition of water (Text S5). In the San Gabriel Mountains, we do not expect the composition of the gneissic bedrock nor the presence of saline groundwater to contribute to changes in resistivity, although we recognize that small-scale changes in salinity may be present between catchments. We assume salinity to be constant through the study period and the changes in resistivity due to the addition or subtraction of water to outweigh effects from salinity in this environment. This is a common assumption in critical zone ERT studies (Cardenas and Kanarek, 2014), and some do not consider the effect of salinity at all (Thayer et al., 2018).

Cardenas, M.B. and Kanarek, M.R., 2014. Soil moisture variation and dynamics across a wildfire burn boundary in a loblolly pine (*Pinus taeda*) forest. *Journal of hydrology*, 519, pp.490-502

Thayer, D., Parsekian, A.D., Hyde, K., Speckman, H., Beverly, D., Ewers, B., Covalt, M., Fantello, N., Kelleners, T., Ohara, N. and Rogers, T., 2018. Geophysical measurements to determine the hydrologic partitioning of snowmelt on a snow-dominated subalpine hillslope. *Water Resources Research*, 54(6), pp.3788-3808.

REVIEWERS' COMMENTS

Reviewer #1 (Remarks to the Author):

The authors have mostly addressed all my concerns from the first review. This is a good paper that describes conceptualization of fire effects to surface and subsurface hydrology. The subsurface contribution of stream response to rain events in fire affected catchments is often ignored and the manuscript helps change that. I would still like to see more on how lessons learned in this study can be applied more broadly, but I also respect that the authors have decided not to extrapolate beyond the chaparral and San Gabriel Mountains. My concern is that this then becomes too incremental for a publication such as Nature Communications – but I believe the conceptual model to be correct for these sites and the evidence provided here supports that conceptualization, which likely applies in many more places.

Reviewer #2 (Remarks to the Author):

The authors response to my review of the earlier submission is good – I am happy that all points (except one minor point – see below) have been addressed. I'm pleased to see the inclusion of the third catchment in the analysis and also some statistical analysis of measured hydraulic conductivities. Having studied the new version and the authors' rebuttal I feel that the manuscript is almost there – I just have a few minor points to raise.

In my review of the original manuscript I commented "The authors say that low Ks [hydraulic conductivity] values in the unburned catchment are probably due to hydrophobic layers due to vegetation. But low values in the burned catchment are due to hydrophobicity from burning. I can understand fire damage leading to hydrophobicity but it seems odd that hydrophobicity is being used to explain behaviour in the two apparently contrasting systems". They did not respond to this comment in the rebuttal. I understand that the authors are giving a reasoning for low Ks value in both unburned and burned systems but I think that they should emphasise that their hypothesis is that they are expected more occurrence of low Ks values in the burned catchments. A short statement to this effect in text S6 would address this.

Line 104-106. The author state "Our findings add to the conceptual understanding of the hydrologic response to storms in post-fire, ravel-dominated small catchments ...". The reference to "ravel" does not appear anywhere in the main manuscript and only appears as an example in the SI (see comment below about text S5), and that is a reference to causes of contact resistance, not processes.

The authors use "time-lapse", "timelapse" and "time lapse" throughout (e.g Line 91, Line 235, Line 392, respectively). Consistency is needed.

Line 388. Brackets needed around date of citation.

Line 673-674 (and text S4). The reference to Blanchy et al. is not appropriate. Rather than using Blanchy et al.(2020a), the authors should use Blanchy et al.(2020b) [see reference details below].

Text S4 states "Time-lapse inversion models were also produced in ResIPy without reciprocal measurements. Here, the ratio of new data to the first dataset is taken and multiplied by a forward model with homogeneous subsurface resistivity, and a sensitivity matrix in calculating percent change in resistivity". If the authors used ResIPy to carry out time-lapse inversion (as stated in the first sentence) then they most likely used a "difference inversion" approach. Their statement of ratioing data and multiplying by a forward model suggests that they used a simpler ratio inversion (which is not a feature of ResIPy, but could be done with some external modification to input files). I think a little clarity is needed.

In text S5 the authors state "A typical threshold for good contact resistance for surface data is 20 kilohms, although this varies with environmental and geological factors", citing Day-Lewis et al

(no date). I have a few issues with this. First, I do not think it is appropriate to cite an unpublished article like this. Second, from my experience 20 kohm is a high contact resistance and I would certainly not consider it "good". It will not necessarily negate the use of data but I disagree with the message as written. Binley and Slater (2020, p121) state "Good contact resistances are on the order of a few kohm or less, although contact resistances in the 10s of kohm range are still usually acceptable for resistivity measurement". And thus I am not saying that a 20 kohm contact resistance makes a measurement unusable, but it is certainly not a "good" contact resistance. The statement by the authors also implies that "environmental and geological factors" will affect the threshold of good/bad. This is true but a critical factor is the input impedance of the instrument being used. The authors may feel that I am being pedantic here (and maybe I am), however, I would not like to see the statement that such a high contact resistance is "good" as it may be quoted by others in the future. And I should note that the contact resistances reported by the authors is much lower than 20 kohm and so my comments do not impact on the analysis within the manuscript.

In text S5. The authors refer to "ravel cones". A simple definition of this may be useful or a reference (e.g., if appropriate, their reference 42 in the main text). The only other reference to ravel in the manuscript is in the abstract (see earlier comment).

REFERENCES

Binley, A. and L. Slater (2020) Resistivity and Induced Polarization. Theory and Applications to the Near-Surface Earth, Cambridge University Press, doi: 10.1017/9781108685955.

Blanchy, G., Watts, C., Richards, J., Bussell, J., Huntenburg, K., Sparkes, D., Stalham, M., Hawkesford, M., Whalley, W., Binley, A. (2020a) Time-lapse geophysical assessment of agricultural practices on soil moisture dynamics, Vadose Zone Journal, 19:e20080, doi: 10.1002/vzj2.20080

Blanchy, G., S. Saneiyani, J. Boyd, P. McLachlan and A. Binley (2020b) ResIPy, an intuitive open source software for complex geoelectrical inversion/modeling in 2D space, Computer & Geosciences, 137, 104423, doi: 10.1016/j.cageo.2020.104423

SUMMARY

I feel that the changes made have strengthened the manuscript and led to a more robust interpretation. I am happy to recommend publication, subject to some very minor changes to the text in response to my comments above.

Andrew Binley
10-Feb-2023

Reviewer #4 (Remarks to the Author):

My review of the manuscript comes after the first round of reviews. I read the paper and the supplementary information prior to reading the response to comments to ensure independence and minimize bias. The study and the manuscript are well-designed. The manuscript was a joy to read and learn from. It is a valuable contribution to the hydrologic literature on wildfire. While I am not actively doing research in this area, the study is a rare combination of the observations they presented. The conclusion is simple and intuitive, in my view, but nonetheless hard to provide clear evidence for. Kudos to the authors for being able to put the evidence together.

The manuscript lays out interesting ideas on the importance of subsurface water flow and storage in response to wildfire clearing of vegetation. The author provided a few circumstantial lines of evidence, which taken together provide compelling support for their conclusions. I find that their observations support their hypotheses.

The initial reviews have provided much to address already. I offer a few technical suggestions below.

On the ER tomograms - Please remove the edges. No data exists there despite the inversion creating something there. Moreover, it would help to show the sensitivity plots. Finally, and this might be more difficult to address, I wonder if temperature is playing a role here. The authors might do and show how soil temp changes would translate to changes in ER, and compare this with those they observed. Could the changes be partly due to different temperature regimes due to difference in sun exposure because of the fire clearing and due to slope and aspect?

L258: Should be Figure 3c

RESPONSE TO REVIEWER COMMENTS

Reviewer #1 (Remarks to the Author) - 4/18/2023:

The authors have mostly addressed all my concerns from the first review. This is a good paper that describes conceptualization of fire effects to surface and subsurface hydrology. The subsurface contribution of stream response to rain events in fire effected catchments is often ignored and the manuscript helps change that. I would still like to see more on how lessons learned in this study can be applied more broadly, but I also respect that the authors have decided not the extrapolate beyond the chaparral and San Gabriel Mountains. My concern is that this then becomes too incremental for a publication such as Nature Communications – but I believe the conceptual model to be correct for these sites and the evidence provided here supports that conceptualization, which likely applies in many more places.

We are glad to hear that the reviewer is happy with our response and edits to the manuscript.

Reviewer #2 (Remarks to the Author) - 4/18/2023:

The authors response to my review of the earlier submission is good – I am happy that all points (except one minor point – see below) have been addressed. I'm pleased to see the inclusion of the third catchment in the analysis and also some statistical analysis of measured hydraulic conductivities. Having studied the new version and the authors' rebuttal I feel that the manuscript is almost there – I just have a few minor points to raise.

In my review of the original manuscript I commented “The authors say that low Ks [hydraulic conductivity] values in the unburned catchment are probably due to hydrophobic layers due to vegetation. But low values in the burned catchment are due to hydrophobicity from burning. I can understand fire damage leading to hydrophobicity but it seems odd that hydrophobicity is being used to explain behaviour in the two apparently contrasting systems”. They did not respond to this comment in the rebuttal. I understand that the authors are giving a reasoning for low Ks value in both unburned and burned systems but I think that they should emphasise that their hypothesis is that they are expected more occurrence of low Ks values in the burned catchments. A short statement to this effect in text S6 would address this.

We thank the reviewer for this comment. We apologize for not addressing this earlier. We have now added a sentence to the supplementary information (Text S7) emphasizing that we expect higher hydrophobicity in the burned catchment, although we do not explicitly see this in the data (Figure 3 in the main text). Text S7 now reads:

“The combined post-fire effect is expected to result in point locations with very high hydrophobicity, as well as some locations with very low hydrophobicity where vegetation debris and duff have been removed.”

Line 104-106. The author state “Our findings add to the conceptual understanding of the hydrologic response to storms in post-fire, ravel-dominated small catchments ...”. The reference to “ravel” does not appear anywhere in the main manuscript and only appears as an example in the SI (see comment below about text S5), and that is a reference to causes of contact resistance, not processes.

We have changed the phrasing of this sentence to read: “...storms in post-fire bedrock catchments...” to align with our description of the San Gabriel Mountains in the Introduction.

The authors use “time-lapse”, “timelapse” and “time lapse” throughout (e.g Line 91, Line 235, Line 392, respectively). Consistency is needed.

We thank the reviewer for catching this detail. We have changed all instances to read “time-lapse”.

Line 388. Brackets needed around date of citation.

The citation has been fixed.

Line 673-674 (and text S4). The reference to Blanchy et al. is not appropriate. Rather than using Blanchy et al.(2020a), the authors should use Blanchy et al.(2020b) [see reference details below].

We have fixed the reference in both the manuscript and the supplement.

Text S4 states “Time-lapse inversion models were also produced in ResIPy without reciprocal measurements. Here, the ratio of new data to the first dataset is taken and multiplied by a forward model with homogeneous subsurface resistivity, and a sensitivity matrix in calculating percent change in resistivity”. If the authors used ResIPy to carry out time-lapse inversion (as stated in the first sentence) then they most likely used a “difference inversion” approach. Their statement of ratioing data and multiplying by a forward model suggests that they used a simpler ratio inversion (which is not a feature of ResIPy, but could be done with some external modification to input files). I think a little clarity is needed.

We thank the reviewer for this clarification. We have edited the text to simply cite the time-lapse inversion default function of ResIPy; we did not employ the use of a forward model.

In text S5 the authors state “A typical threshold for good contact resistance for surface data is 20 kilohms, although this varies with environmental and geological factors”, citing Day-Lewis et al (no date). I have a few issues with this. First, I do not think it is appropriate to cite an unpublished article like this. Second, from my experience 20 kohm is a high contact resistance and I would certainly not consider it “good”. It will not necessarily negate the use of data but I disagree with the message as written. Binley and Slater (2020, p121) state “Good contact resistances are on the order of a few kohm or less, although contact resistances in the 10s of

kohm range are still usually acceptable for resistivity measurement". And thus I am not saying that a 20 kohm contact resistance makes a measurement unusable, but it is certainly not a "good" contact resistance. The statement by the authors also implies that "environmental and geological factors" will affect the threshold of good/bad. This is true but a critical factor is the input impedance of the instrument being used. The authors may feel that I am being pedantic here (and maybe I am), however, I would not like to see the statement that such a high contact resistance is "good" as it may be quoted by others in the future. And I should note that the contact resistances reported by the authors is much lower than 20 kohm and so my comments do not impact on the analysis within the manuscript.

We thank the reviewer for clarification on this subject. We have removed the reference as well as any specific threshold for contact resistance, as we report the ranges in Table S5. The contact resistances reported in the supplement were incorrectly labeled with the unit "ohm"; we have corrected this typo to read "kohm". Considering this change, many of our surveys include contact resistances that hover within the 10s of kohm range, which should not impact our analysis given the general guidance from the reviewer and the provided reference.

In text S5. The authors refer to "ravel cones". A simple definition of this may be useful or a reference (e.g., if appropriate, their reference 42 in the main text). The only other reference to ravel in the manuscript is in the abstract (see earlier comment).

We have removed the word "ravel" in favor of "gravelly soils", which we previously had listed in the text.

REFERENCES

Binley, A. and L. Slater (2020) Resistivity and Induced Polarization. Theory and Applications to the Near-Surface Earth, Cambridge University Press, doi: 10.1017/9781108685955.

Blanchy, G., Watts, C., Richards, J., Bussell, J., Huntenburg, K., Sparkes, D., Stalham, M., Hawkesford, M., Whalley, W., Binley, A. (2020a) Time-lapse geophysical assessment of agricultural practices on soil moisture dynamics, Vadose Zone Journal, 19:e20080, doi: 10.1002/vzj2.20080

Blanchy, G., S. Saneiyani, J. Boyd, P. McLachlan and A. Binley (2020b) ResIPy, an intuitive open source software for complex geoelectrical inversion/modeling in 2D space, Computer & Geosciences, 137, 104423, doi: 10.1016/j.cageo.2020.104423

SUMMARY

I feel that the changes made have strengthened the manuscript and led to a more robust interpretation. I am happy to recommend publication, subject to some very minor changes to the text in response to my comments above.

We are pleased that the Reviewer finds our interpretation more robust following revisions, and thank them for the constructive and thorough reviews.

Andrew Binley
10-Feb-2023

Reviewer #4 (Remarks to the Author) - 4/18/2022:

My review of the manuscript comes after the first round of reviews. I read the paper and the supplementary information prior to reading the response to comments to ensure independence and minimize bias. The study and the manuscript are well-designed. The manuscript was a joy to read and learn from. It is a valuable contribution to the hydrologic literature on wildfire. While I am not actively doing research in this area, the study is a rare combination of the observations they presented. The conclusion is simple and intuitive, in my view, but nonetheless hard to provide clear evidence for. Kudos to the authors for being able to put the evidence together.

The manuscript lays out interesting ideas on the importance of subsurface water flow and storage in response to wildfire clearing of vegetation. The author provided a few circumstantial lines of evidence, which taken together provide compelling support for their conclusions. I find that their observations support their hypotheses.

The initial reviews have provided much to address already. I offer a few technical suggestions below.

On the ER tomograms - Please remove the edges. No data exists there despite the inversion creating something there. Moreover, it would help to show the sensitivity plots. Finally, and this might be more difficult to address, I wonder if temperature is playing a role here. The authors might do and show how soil temp changes would translate to changes in ER, and compare this with those they observed. Could the changes be partly due to different temperature regimes due to difference in sun exposure because of the fire clearing and due to slope and aspect?

We thank the reviewer for the thoughtful comments and feedback. We have removed the edges from the resistivity models in Figure 4. Rather than show the sensitivity plots, we have included an example of the depth of investigation (DOI) at catchment Thelma from the initial December 2020 survey to illustrate the depth limits of our models (Figure S10).

We collected air surface temperature data over the period of December 202 through early February 2021 at the mouth of the catchment Thelma. We have examined the data and noted average diurnal temperature swings of ~3 degrees C, which corresponds to ~14% change in resistivity within the top ~1 m of our surveys. This change is minor when compared to the post-storm changes we observe in the ERI models during winter months, particularly at depth. Further details have been included in the supplementary information (Text S5). Temperature likely contributed partially to our data, but we do not consider it the main driver of the observed resistivity changes.

L258: Should be Figure 3c

We have removed the figure reference from this sentence because we are hypothesizing and not explaining our data. The figure reference was confusing.